

# Physico-chemical and biological factors influencing dinoflagellate cyst production in the Cariaco Basin

Manuel Bringué[1,2], Robert C. Thunell[1], Vera Pospelova[2], James L. Pinckney[1,3], Oscar E. Romero[4], Eric J. Tappa[1]

[1] School of the Earth, Ocean and Environment, University of South Carolina, 701 Sumter Street, EWS 617, Columbia, SC 29208, USA
[2] School of Earth and Ocean Sciences, University of Victoria, PO Box 1700, STN CSC, Victoria, BC, Canada V8W 2Y2
[3] Belle W. Baruch Institute for Marine and Coastal Sciences, University of South Carolina, 700 Sumter Street, EWS 604, Columbia, SC 29208, USA
[4] MARUM, Center for Marine Environmental Sciences, University Bremen, Leobenerstraße, 28359 Bremen, Germany

*Correspondence to*: Manuel Bringué (mbringue@geol.sc.edu)

**Abstract.** We present a 2.5 year-long sediment trap record of dinoflagellate cyst production in the Cariaco Basin, off Venezuela (southern Caribbean Sea). The site lies under the influence of wind-driven, seasonal upwelling which promotes high levels of primary productivity during boreal winter and spring. Changes in dinoflagellate cyst production is documented between November 1996 and May 1999 at ~ 14 day intervals and interpreted in the context of *in situ* observations of physico-chemical and biological parameters measured at the mooring site.

Dinoflagellate cyst assemblages are diverse (57 taxa) and dominated by cyst taxa of heterotrophic affinity, primarily *Brigantedinium* spp. (51% of the total trap assemblage). Average cyst fluxes to the trap are high ($17.1 \times 10^3$ cysts m$^{-2}$ day$^{-1}$) and show great seasonal and interannual variability. On seasonal time scales, dinoflagellate cyst production responds closely to variations in upwelling strength, with increases in cyst fluxes of several protoperidinioid taxa observed during active upwelling intervals, predominantly *Brigantedinium* spp. Cyst taxa produced by autotrophic dinoflagellates, in particular *Bitectatodinium spongium*, also respond positively to upwelling. Several 'spiny brown' cysts contribute substantially to the assemblages, including *Echinidinium delicatum* (9.7%) and *Echinidinium granulatum* (7.3%), and show a closer affinity to weaker upwelling conditions. The strong El Niño event of 1997/98 appears to have negatively impacted cyst production in the basin with a one year lag, and may have contributed to the unusually high fluxes of cysts type 'Cp' (possibly the cysts of the toxic dinoflagellate *Cochlodinium polykrikoides* sensu Li et al. 2015), with cyst type Cp fluxes up to $11.8 \times 10^3$ cysts m$^{-2}$ day$^{-1}$ observed during the weak upwelling event of Feb.-May 1999.

Possible trophic interactions between dinoflagellates and other major planktonic groups are also investigated by comparing the timing and magnitude of cyst production with proxies for phytoplanktonic communities (from photopigment data) and micro- to macrozooplankton abundance indicators (from palynological data) at the site. This work provides new, detailed insights into the ecology of cyst-producing dinoflagellates and will allow for more detailed interpretations of fossil assemblages extracted from sedimentary records in the basin and elsewhere.



**Keywords**

Upwelling, primary productivity, sediment trap, Harmful Algal Blooms, southern Caribbean Sea

## 1 Introduction

The Cariaco Basin, located off the coast of Venezuela, harbors one of the most productive ecosystems in the Caribbean Sea (e.g., Müller-Karger and Castro, 1994; Antoine et al., 1996; Rueda-Roa and Müller-Karger, 2013). Seasonal, trade wind-driven upwelling promotes high levels of primary productivity in the southern Caribbean upwelling system, typically during boreal winter and spring, with secondary upwelling events often observed in the basin in July and August (e.g., Müller-Karger and Castro, 1994; Astor et al., 2003). Microbial respiration of sinking organic matter, together with restricted bottom

water ventilation in the basin, have led to the formation of anoxic conditions below ~ 275 m (Richards and Vaccaro, 1956; Deuser, 1973; Astor et al., 2003). The succession of high fluxes of biogenous components (carbonate, opal and organic matter) during active upwelling intervals and dominantly lithogenous material during the rainy season (Thunell et al., 2000; Goñi et al., 2003) is well reflected in the sediments at the site (Peterson et al., 1991; Hughen et al., 1996). In turn, the sedimentary record of the basin is widely used as a repository of past climate and environmental change in the region (e.g.,

Hughen et al., 1996, 2000; Black et al., 1999, 2007; Peterson et al., 1991, 2000).

Station CARIACO has been the site of high frequency hydrographic and biogeochemical monitoring since 1995, with monthly water column sampling and continuous deployment of sediment traps conducted as part of the Cariaco Ocean Time-Series Program (Thunell et al., 2000; Müller-Karger et al., 2001). This United States–Venezuelan research effort constitutes a unique opportunity to investigate biogeochemical and ecological processes that take place in the water column, their

temporal evolution, and how these processes are encoded in the settling flux of particles (e.g., Ho et al., 2002; Astor et al., 2003; Goñi et al., 2003; Taylor et al., 2012; Marshall et al., 2013; Montes et al., 2013; Calvert et al., 2015). We present the first study to focus on dinoflagellate cyst production in the Caribbean Sea, using sediment trap samples collected over a 2.5 year period at station CARIACO.

Dinoflagellates are aquatic protists that form an important component of marine plankton in terms of diversity and biomass,

both as primary producers and as heterotrophic grazers on a wide range of phyto- and zooplanktonic organisms (e.g., Taylor, 1987; Jacobson and Anderson, 1996; Jeong, 1999; Calbet, 2008). In addition to their key role in coastal marine food webs, dinoflagellates are a valuable tool to investigate the history of changes in sea-surface conditions and planktonic communities (e.g., Dale, 1996; de Vernal et al., 2001; Harland et al., 2013; Bringué et al., 2014; Pospelova et al., 2015; Ellegaard et al., 2017), since many dinoflagellates produce resting cysts as part of their life cycle that are well preserved in fine-grained

sediments over geological times. Establishing the detailed environmental preferences and controls over cyst-producing dinoflagellates has been the focus of many studies over the last few decades, and has traditionally been tackled by studying the distribution of cysts in 'surface' sediments in relation to (multi-)annually averaged sea-surface conditions (e.g., Rochon





et al., 1999; Zonneveld et al., 2013, and references therein). Alternatively, sediment trap records can resolve the *seasonality* in cyst production and thus considerably improve the accuracy of environmental signals associated with each dinoflagellate species (e.g., Dale, 1992; Dale and Dale, 1992; Morquecho and Lechuga-Deveze, 2004; Susek et al., 2005; Fujii and Matsuoka, 2006; Pitcher and Joyce, 2009; Howe et al., 2010; Pospelova et al., 2010, in press; Zonneveld et al., 2010; Price and Pospelova, 2011; Bringué et al., 2013; Prebble et al., 2013; Heikkilä et al., 2016).

Since the early work of Wall (1967) on five sub-samples of a core from the Cariaco 'Trench', the dinoflagellate cyst sedimentary record of the basin has been studied by González et al. (2008; ~73 to 30 ka interval) and Mertens et al. (2009a; last 30 ka) at multi-centennial to millennial resolution. This study focusses on seasonal cyst production as a calibration effort for the interpretation of fossil assemblages from the basin's pristine sedimentary record and elsewhere.

10 Our objectives are: 1- to document the seasonal changes in dinoflagellate cyst production in the basin; 2- to relate cyst production to physico-chemical parameters (e.g., temperature, salinity, nutrients); and 3- to investigate the relationships between dinoflagellate species and other major planktonic groups (e.g., diatoms, haptophytes, copepods) based on *in situ* observations.

## 2 Environmental setting

15 The Cariaco Basin is a 1400 m deep depression on the continental shelf off Venezuela in the southern Caribbean Sea (Fig. 1). The western and eastern sub-basins, separated by an ~ 900 m deep saddle, are partially isolated from the Caribbean and Atlantic waters by a shallow (~100 m) ridge on its northern flank that is only cut by two small channels: Canal Centinela (146 m) and Canal de la Tortuga (135 m).

Surface waters originate in the North Equatorial and Guyana Currents and enter the Caribbean Sea from the southeast 20 (Gordon, 1967; Richards, 1975). Waters from the Amazon and Orinoco Rivers that may be transported into the Caribbean have little influence on the hydrography of the basin (Müller-Karger et al., 1989). Below sill depth, the combination of restricted water circulation, a steep density gradient over the upper 150 m and a high flux of organic matter settling in the basin results in slow renewal of deep waters and anoxic conditions below ~275 m (Richards and Vaccaro, 1956; Deuser, 1973; Thunell et al., 2000; Astor et al., 2003).

25 Primary production in the basin is high (> 300 g C $m^{-2}$ $yr^{-1}$) and shows great seasonal and interannual variability (e.g., Müller-Karger et al., 2001, 2004). Seasonal upwelling of Subtropical Underwater (SUW), a water mass characterized by a salinity of 36.9 and a nitrate concentration of 5–10 μM (Morrison and Smith, 1990), is controlled by the position of the Intertropical Convergence Zone (ITCZ). When the ITCZ lies south of the equator (typically from December to April), strong E–NE trade winds (> 6 m $s^{-1}$) induce wind-driven upwelling along the coastline, bringing nutrients to the surface and 30 fostering high levels of primary productivity (e.g., Richards, 1975; Müller-Karger et al., 2001). During the summer/fall rainy season, the ITCZ migrates to its northern position, winds weaken and upwelling ceases, although shorter, secondary upwelling events are often observed in July and August (e.g., Astor et al., 2003; Goñi et al., 2003). Interannual variability in



primary productivity at the CARIACO station (eastern basin) is also pronounced, with annually-integrated measurements between 1996 and 2001 ranging from ~370 to 650 g C m$^{-2}$ yr$^{-1}$ (Müller-Karger et al., 2004). Surface salinity at the CARIACO station varies from > 36.8 in January–July to < 36.6 during the rainy season (Astor et al., 2003). The depth of the euphotic zone (defined as the depth of 1% photosynthetic active radiation level), measured between 1995 and 2005, was

usually shallower under active upwelling conditions (36.7 ± 12.3 m) than during the rainy season (47.9 ± 13.5 m; Lorenzoni et al., 2011).

Phases of the North Atlantic Oscillation (NAO), defined as the difference in sea level pressure anomalies between the Azores High and the Icelandic Low, are known to modulate precipitation in the Caribbean (Giannini et al., 2001) but appear to have limited impact on SST patterns and hydrography of the basin (Taylor et al., 2012; Astor et al., 2013). The El Niño–

Southern Oscillation (ENSO) phenomenon, which affects a major portion of the tropical Pacific, the Americas and the Caribbean through atmospheric teleconnections, has been shown to influence precipitation, SST and productivity patterns at the CARIACO station with a one year lag (Enfield and Mayer, 1997; Giannini et al., 2001; Taylor et al., 2012; Astor et al., 2013), and may cause shifts in phytoplankton communities in the basin (Goñi et al., 2003; Romero et al., 2009).

The high levels of primary productivity result in the export of large fluxes of particulate organic matter to the depths

(Thunell et al., 2000, 2007), while local rivers (Tuy, Unare, and Neveri) deliver the bulk of terrigenous sediments to the basin (e.g., Elmore et al., 2009; Bout-Roumazeilles et al., 2013). The succession of intervals of high surface productivity during active upwelling (and associated flux of biogenous material), and periods of low productivity and rainy conditions when sedimentation is dominated by lithogenous particles, is reflected in the laminated sediments that accumulate under anoxic conditions in the basin (e.g., Peterson et al., 1991; Hughen et al., 1996; Thunell et al., 2000; Goñi et al., 2003).

## 3 Material and methods

### 3.1 Sample collection and analyses

All samples used in this study were collected as part of the Cariaco Ocean Time-Series Program at a station located in the eastern basin (10°30′ N, 64°40′ W; Fig. 1). The program has been collecting oceanographic observations since 1995, with monthly water column sampling and nearly continuous deployment of sediment traps at the CARIACO station (Thunell et

al., 2000; Müller-Karger et al., 2001). Conductivity-Temperature-Depth (CTD) casts (providing SST, SSS and $\sigma_T$ profiles) and water column sampling at discrete depths using a rosette equipped with 12 Teflon-coated Niskin bottles (for measurements of various parameters including dissolved oxygen, pH, alkalinity, chlorophyll *a* concentrations, rates of primary productivity, suspended particulate organic carbon and nitrogen, phaeopigments and nutrients) were performed on a monthly basis. An overview of the program and methods is provided in Müller-Karger et al. (2001), and sampling, analytical

procedures and data can be found at www.imars.usf.edu/cariaco.

Four Mark-VII automated sediment traps (Honjo and Doherty, 1988) were deployed on a single mooring at depths of 275 m (Trap A), 455 m (B), 930 m (C) and 1255 m (D) at the CARIACO station, as described in Thunell et al. (2000). The conical



traps have an aperture of 0.5 m² and are equipped with a baffle to reduce turbulence over the collection area. Thirteen cups filled with a buffered formalin solution (as a preservative for the organic matter) and mounted on a rotating carousel collected samples at 2 week intervals, except for a few cups that collected sediments for 7–12 days. The mooring was recovered and redeployed every 6 months. The present study focusses on samples from Trap A (275 m) collected during

deployments 3 to 7, i.e., between November 8, 1996 and May 3, 1999.

Upon recovery, sediment trap samples were stored in sealed containers and kept refrigerated in the dark. Whole trap samples were split using a precision rotary splitter and swimmers that are not part of the particle flux were removed before analyses. Quarter splits were thoroughly rinsed, oven- (50°C) or freeze-dried, weighed and used for geochemical analyses. Contents of particulate organic carbon (thereafter $C_{org}$) and particulate organic nitrogen (N) were measured using a Perkin-Elmer 2400

elemental analyzer; biogenic silica (bioSi) was analysed following the method of Mortlock and Froelich (1989) and carbonate ($CaCO_3$) content was determined using the method described in Osterman et al. (1990) (Thunell et al., 2000, 2007). Sixteenth (1/16) splits of whole trap samples were used for palynological analyses, as described below (Sect. 3.2).

This study uses some previously published data, for means of interpretation of dinoflagellate cyst data in the context of other important planktonic groups in the basin. In particular, diatom and silicoflagellate census data, determined from different

splits of the same sediment trap samples, were obtained from Romero et al. (2009), and high performance liquid chromatography (HPLC) measurements of photopigments, analysed from monthly water column sampling at the CARIACO station, were obtained from Pinckney et al. (2015).

### 3.2 Palynological preparation

The 62 sediment trap samples, representing 1/16 splits of all Trap A (275 m) cups from deployments 3 through 7, were

processed in the Paleoenvironmental/Marine Palynology Laboratory at the University of Victoria (BC, Canada). Dinoflagellate cysts and other palynomorphs (including pollen grains and spores, microforaminiferal organic linings, copepod eggs and dinoflagellate thecae) were extracted using the palynological processing technique described by Pospelova et al. (2005, 2010) to ensure optimal recovery of all palynomorphs. Sediment trap samples were rinsed 3 times to remove salt residue, oven dried at 40 °C and weighed with an analytical balance. One calibrated tablet of *Lycopodium clavatum* (batch

no. 3862, produced and distributed by the Department of Quaternary Geology, University of Lund, Sweden) was added to each sample in order to estimate palynomorph concentrations (Stockmarr, 1971; Mertens et al., 2009b, 2012). Samples were treated with room temperature 10% HCl to remove carbonates. Particles finer than 15 μm and coarser than 120 μm were eliminated by wet sieving through Nitex nylon meshes. Samples were then exposed to room temperature 48% HF for 2–3 days to remove silicates, followed by a second 10% HCl treatment to eliminate precipitated fluorosilicates. After each step,

samples were rinsed with reversed osmosis water and centrifuged at 3600 rpm for six minutes. Samples were gently sonicated for up to a minute (Price et al., 2016) and collected on a 15 μm mesh. One or two drops of residue were mounted in glycerine jelly between a slide and cover slip.





Dinoflagellate cysts and other palynomorphs were identified and counted using Nikon Eclipse transmitting light microscopes (models E200 and 80*i*) at 500×, 600× and 1000× magnifications. A minimum of 300 cysts per sample were counted, except in 6 cases where the splits did not contain enough cysts or when that number could not be reached after counting two full slides. In two instances, pairs of adjacent samples representing only 7 days of sediment collection each (and yielding very little material) were combined together, and each couplet was treated as one sample of 14 days, with flux data for individual taxa calculated as an average of the fluxes in the two 7-day samples (Table S1). Overall, between 126 and 460 cysts were counted per sample, with an average of 316. Palynomorph fluxes are expressed as specimens per square meter per day.

Error on cyst fluxes was estimated as in Bringué et al. (2014), by adding the systematic error in flux estimates associated with the *L. clavatum* tablets (9.82%; Maher, 1981), and the statistical error in counts (10.1%, considering both *L. clavatum* and dinoflagellate cyst counts) in quadrature, resulting in a total error on cyst fluxes estimates of 14.1%. All samples and slides are stored at the paleoenvironmental/Marine Palynology Laboratory, School of Earth and Ocean Sciences, University of Victoria, Canada.

**3.3 Dinoflagellate cyst nomenclature**

The dinoflagellate cyst nomenclature follows the paleontological taxonomy system provided in Lentin and Williams (1993) (recently updated in Williams et al., 2017) and conforms to taxonomic descriptions provided by Margalef (1961), Wall and Dale (1966, 1968, 1969), Reid (1977), Matsuoka et al. (1990, 2006, 2009), Zonneveld (1997), Ellegaard and Moestrup (1999), Head (1996, 2002), Rochon et al. (1999), Zonneveld and Jurkschat (1999), Verleye et al. (2011), Mertens et al. (2013) and Liu et al. (2015), with most cyst description summaries and corresponding motile dinoflagellate names provided in Zonneveld and Pospelova (2015).

The cysts of cf. *Biecheleria* sp., *Echinidinium* cf. *delicatum* and cyst type M1 were identified as in Bringué et al. (2014, 2016). The cysts of cf. *Ensiculfera carinata* (identified after dissolution of the calcareous outer layer by HCl treatment) differs from the description provided by Matsuoka et al. (1990) by the brown color of the inner organic layer. The cysts of *Diplopelta* cf. *symmetrica* conform to the description of Dale et al. 1993 except for the central body diameter that can be as big as 57 μm in some specimens. The cysts of cf. *Diplopelta* sp. are spherical (30–37 μm in diameter), brown cysts that bear numerous, hair-like, hollow processes between 2 and 5.6 μm in length. *Dubridinium* cf. *ulsterum* corresponds to the description provided by Reid (1977) but is only slightly smaller in size (max. diameter of 44 μm). 'Cyst type Cp', labelled 'Cp' due to their resemblance with the cysts of *Cochlodinium polykrikoides* sensu Li et al. 2015, are large, subspherical to elongated cysts (max. height of up to 55 μm) with a thin, light brown cyst wall bearing ~ 2 μm-high reticulate ornaments of a slightly darker brown shade. Cysts type Cp are larger and more elongated than *C. polykrikoides* cysts sensu Li et al. 2015.

'Spiny brown type A' cysts (hereafter SBA) are relatively small, spherical (25–30 μm diameter) chorate cysts with smooth to microgranulated wall and numerous, long (7–11 μm), apiculocavate, capitate processes. SBA may correspond to the cysts of *Archaeperidinium bailongense* reported from Pacific waters (Liu et al., 2015) but has a smaller central body diameter and a higher process length to cyst body diameter ratio. 'Spiny brown type C' (hereafter SBC) has a spherical central body (25–34





μm diameter) with clearly visible microgranulations on the cyst wall; processes are numerous, 5 to 8 μm in length, hollow, randomly distributed, and taper from a relatively large base (up to 2.5 μm) to an acuminate tip. SBC may be a morphotype of *Echinidinium granulatum* but was counted and analysed separately because of differences in process shape and trends in the timing of cyst production. 'Spiny brown type D' (SBD) is a small (23–27 μm diameter), light brown cyst with a smooth wall

and sparse, thin, usually erect but often bent, hollow and acuminate processes 5–8 μm in length. Cysts of *Protoperidinium* sp. C are spherical, pale brown cysts with a loose, thin periphragm of variable height (max. of 4–7 μm) surrounding the central body, and attached to the endophragm by numerous folds arranged in meandering lines. Cysts of *Protoperidinium* sp. C resemble the cysts of *Protoperidinium parthenopes* (Kawami and Matsuoka, 2009) but have a larger central body diameter (34–45 μm), and thinner and more numerous folds connecting the two wall layers.

Several cyst taxa were grouped due to morphological similarities. In particular, *Spiniferites bulloideus* is grouped with *Spiniferites ramosus*, and *Spiniferites hyperacanthus* is grouped with *Spiniferites mirabilis*. *Dubridinium caperatum* and *D.* cf. *ulsterum* are grouped with *Dubridinium* spp.*, Echinidinium* cf. *delicatum* is grouped with *Echinidinium delicatum*, the cysts of *Protoperidinium nudum* are grouped with *Selenopemphix quanta*. Both chordate and rhombic morphotypes of the cysts of *Protoperidinium oblongum*, together with *Votadinium calvum*, were grouped as the cysts of *P. oblongum*.

*Brigantedinium* spp. include *B. cariacoense*, *B. irregulare*, *B. majusculum* and *B. simplex* as well as other smooth round brown cysts since archeopyles were not always visible due to unfavourable orientations or folding. *Operculodinium centrocarpum* sensu Wall and Dale 1966 includes both the morphotype described by Wall and Dale (1966) and the morphotype with "reduced processes". All dinoflagellate cysts identified in this study are listed in Table 1, along with their motile equivalents. Note that the expressions "autotrophic cyst" and "heterotrophic cyst" in the text refer to cysts produced

by autotrophic and heterotrophic dinoflagellates, respectively.

**3.4 Statistical analyses**

Multivariate analyses were used to investigate the relationships between individual dinoflagellate cyst taxa and 1- physico-chemical, and 2- biological parameters. Detrended canonical correspondence analyses (DCCA) and Redundancy Analyses (RDA) were performed on dinoflagellate cyst fluxes using CANOCO 4.5 for Windows (ter Braak and Šmilauer, 2002). Both

types are direct gradient analyses, meaning that the ordination of 'species' data (individual cyst taxa) in the multivariate space is further constrained to be a linear combination of the environmental (physico-chemical or biological) variables. Cyst taxa that never contributed more than 1.5% to the assemblages were excluded from the multivariate analyses to reduce noise. DCCA were used first to determine the character of variability within the 'species' data (cyst assemblages). A first DCCA of untransformed cyst data with physico-chemical parameters indicated a length of the first gradient of 2.34 standard deviations

(sd). A second DCCA of square-root transformed cyst data with biological parameters resulted in a length of the first gradient of 1.76 sd. Being far smaller than 4 sd in both cases, the gradients indicate a linear variation in the cyst data and justify the further use of RDA (Lepš and Šmilauer, 2003). Statistical significance of each environmental variable and



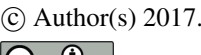

ordination axis was determined using Monte Carlo tests based on 9999 permutations restricted for a time series data structure (Lepš and Šmilauer, 2003).

Physico-chemical variables were obtained primarily from the CARIACO program repository (www.imars.usf.edu/cariaco). Measurement of water column parameters at 25 m depth (SST, SSS, $\sigma_T$, concentrations of $PO_4$, $Si(OH)_4$ and phaeopigments,

C/N ratio of particulate organic matter), from monthly CTD casts and rosette sampling at the mooring site, were selected for the analyses. Chlorophyll $a$ (Chl $a$) measurements determined from rosette casts were integrated over the upper 100 m of the water column to better reflect primary production at the site. The mixed layer depth (MLD) is defined as the depth where the change in density > 0.125 kg m$^{-3}$ (as in Pinckney et al., 2015; C. Benitez-Nelson, pers. comm.). Contents of bioSi, $C_{org}$, N and $CaCO_3$ refer to the parameters measured directly on sediment trap samples. Daily precipitation data were acquired from

the GPCP Version 1.2 One-Degree Daily Precipitation Data Set (data set ds728.3, available from https://rda.ucar.edu; Huffman et al., 2016) and consist in daily precipitation estimates for two 1° latitude × 1° longitude boxes centered on 10.5°N/65.5°W and 10.5°N/64.5°W (therefore covering all the Cariaco Basin area).

Biological variables, representing estimates for abundances of various planktonic groups, were determined from palynological data (this study), diatom and silicoflagellate census data from Romero et al. (2009), and HPLC photopigment

data from Pinckney et al. (2015). In particular, fluxes of invertebrate mandibles, copepod eggs, ciliates and microforaminiferal organic linings are derived from our palynological analyses. All other biological variables used in RDA (namely prasinophytes, cyanobacteria, cryptophytes, haptophytes, chlorophytes, diatoms and 'autotrophic dinoflagellates') are derived from HPLC pigment data as determined in Pinckney et al. (2015) and were integrated over the upper 100 m. All physico-chemical and biological variables used in multivariate analyses are provided in Supplementary material (Table S2).

## 4 Results

### 4.1 Evolution of sea-surface parameters during the study period

The seasonal cycle of upwelling that brings colder, nutrient rich waters to the surface and promotes primary productivity is well illustrated in the records of SST, $\sigma_T$, $PO_4$ and Chl $a$ during the study period (Fig. 2). In order to better characterise the variability in upwelling state and associated biological response, we define intervals of active (weak, moderate and strong)

and relaxed upwelling based on which isotherm reaches a depth of 50 m, determined from SST measurements at the study site (Fig. 2A). Conditions of 'weak', 'moderate' and 'strong' upwelling correspond to intervals when the 22, 21 and 20°C isotherms reach 50 m depth, respectively. All other intervals (when SSTs at 50 m are > 22°C) are considered to represent 'relaxed' upwelling conditions. This practical definition is supported by previous observations in the basin (e.g., Müller-Karger et al., 2001; Astor et al., 2003; Goñi et al., 2003; Taylor et al., 2012).

During the 2.5 years of the time series, SSTs in the upper 100 m of the water column varied between ~19 and 28°C, with lower temperatures at depth and generally during the upwelling season, and higher temperatures in shallow waters during upwelling relaxation (Fig. 2A). The time series started with the end of upwelling relaxation of 1996, followed by a long



interval of active upwelling conditions (January to May 1997) and a secondary upwelling event in July-August 1997. The upwelling was strongest in March 1997, leading to high levels of Chl *a* sustained through May of that year (Fig. 2D). The following upwelling cycle started at the end of November 1997 and is more easily described as two 'pulses' of weak to moderate upwelling ending in April 1998, with an additional, shorter weak upwelling interval in July 1998. After an ~ six

month-long upwelling relaxation, weak upwelling conditions were observed again from mid-January to April 1999.

With sea surface salinity (SSS) values varying only slightly (range of 36.5 to 37.0 over the top 100 m; not shown), temperature was the primary driver of density change (Fig. 2B), with the lowest $\sigma_T$ values (< 24.6 kg m$^{-3}$) at the surface and strongest stratification (compressed isopycnals) observed along the warmest intervals in the record (November–December 1996, September–November 1997 and August–December 1998).

When nutrients are available above the mixed layer depth (see the example of phosphates in Fig. 2C), primary productivity is stimulated and elevated concentrations of Chl *a* are observed (Fig. 2D). This is the case during most active upwelling intervals, with the exceptions of the July–August 1997 upwelling event that apparently yielded very little Chl *a* increase, and a few deep chlorophyll maxima, notably in June 1998 and April 1999.

**4.2 Main fluxes to the trap**

The main biogenic components of the sediment flux to the trap are shown in Fig. 3. Our time series spans ~2.5 years and is only interrupted by small gaps due to short lags between trap recovery and redeployments, and high primary productivity rates that caused the clogging of the trap in April and May 1997 (Goñi et al., 2003). Fluxes of biogenic material share the general pattern of total mass flux (Fig. 3A) but follow more closely the upwelling cycle than the total mass flux, which is also supplied by terrigenous material during the rainy (summer/fall) seasons. Mass fluxes of bioSi, CaCO$_3$ and C$_{org}$ generally

increase under active upwelling conditions, with the highest values recorded during the upwelling events of 1997 and 1998 (up to 0.32 g m$^{-2}$ day$^{-1}$ for bioSi and CaCO$_3$, and 0.18 g m$^{-2}$ day$^{-1}$ for C$_{org}$). However, the weak upwelling of January–April 1999 constitutes a notable exception to this pattern, with some of the lowest fluxes of biogenic material in our record (Fig. 3A).

Diatom and silicoflagellate fluxes (data from Romero et al., 2009; Fig. 3B) show very low abundances during upwelling

relaxation. The record also shows only modest contributions during active upwelling intervals of 1996, 1997 and 1998. In contrast, the weak upwelling event of 1999 is marked by the highest fluxes of diatom valves and silicoflagellate skeletons of the record (up to 21.5 and 2.4 × 10$^6$ individuals m$^{-2}$ day$^{-1}$, respectively).

The most abundant palynomorphs in the time series are ciliates, dinoflagellate cysts, invertebrate mandibles, motile dinoflagellates, copepod eggs, microforaminiferal organic linings and pollen & spores, with average fluxes of 18.6, 17.1,

14.1, 13.5, 3.4, 1.8 and 0.5 × 10$^3$ specimens m$^{-2}$ day$^{-1}$, respectively (Fig. 3C). Abundances of dinoflagellate cysts, overwhelmingly of heterotrophic affinity, vary between 1.6 × 10$^3$ cysts m$^{-2}$ day$^{-1}$ during upwelling relaxation, and 69.2 × 10$^3$ cysts m$^{-2}$ day$^{-1}$ under conditions of active upwelling. In fact, variations in cyst fluxes are tightly coupled with the state of upwelling, as described in more detail below (Sect. 4.3). Abundant motile dinoflagellates were also recovered in the





sediment trap samples, and are reported in Fig. 3C at the genus or family level. Our record indicates that *Dinophysis*/*Phalacroma* species consistently occur under active upwelling conditions, and even more prominently, *Prorocentrum* species (primarily *P. compressum*, *P. micans* and *P. gracile*) can form very large blooms, as observed in January-February 1998 (up to $183.4 \times 10^3$ cysts m$^{-2}$ day$^{-1}$).

Mandibles and copepod eggs, both indicators of zooplankton abundance, show generally higher fluxes during active upwelling intervals, with peak values recorded following moderate to strong upwelling events (Fig. 3C). Ciliates also seem to respond positively to upwelling but show markedly increased abundances during upwelling relaxation as well, especially in the summer and fall of 1998. Fluxes of microforaminiferal organic linings are generally under $2 \times 10^3$ linings m$^{-2}$ day$^{-1}$, except for three 2–4 month-long intervals, two of which occurred under relaxed upwelling conditions (summer/fall of 1997

and 1998). Pollen grains and spores, although routinely observed, rarely exceed $1 \times 10^3$ grains m$^{-2}$ day$^{-1}$ and thus from only a minor component of the biogenic flux, which is consistent with previous studies that showed that the organic matter delivered to the shallow traps at the CARIACO station is overwhelmingly marine (e.g., Thunell et al., 2000; Goñi et al., 2003).

**4.3 Dinoflagellate cyst assemblages and fluxes**

Over the duration of the time series, the total trap assemblage is dominated by *Brigantedinium* spp. (51.0%), accompanied by *Echinidinium delicatum* (9.7%), *Echinidinium granulatum* (7.3%), SBC (5.7%), cyst type Cp (4.6%) and SBA (3.7%). All other cyst taxa account for less than 3% of the total assemblage (Fig. 4A).

While *Brigantedinium* spp. dominate most assemblages in the time series (Fig. 4C), some intervals show a greater contribution *Echinidinium* species and other spiny brown cysts (especially from Jun. 1997 to Jan. 1998), the cysts of cf.

*Ensiculifera carinata* (mainly in Jun.–Oct. 1997) and gymnodiniales (primarily cysts type Cp in Feb.-May 1999).

Seasonal variations in dinoflagellate cyst production are illustrated in Fig. 5. Overall, most cyst taxa respond positively to upwelling strength, but some taxa in particular clearly show increased fluxes during active upwelling intervals. Of those, *Brigantedinium* spp. show the closest and more consistent response to upwelling strength, with higher fluxes (up to $35.2 \times 10^3$ cysts m$^{-2}$ day$^{-1}$) recorded during all active upwelling intervals. The upwelling event of Jan.–May 1997, the strongest in

our record, resulted in the most pronounced increases in *Brigantedinium* spp., SBC, cysts of cf. *Diplopelta* sp., *Dubridinium* spp., *Quinquecuspis concreta*, *Echinidinium* spp., *Selenopemphix quanta* and *Selenopemphix nephroides* (Fig. 5). The latter two taxa (*S. quanta* and *S. nephroides*) also show high abundances during the secondary upwelling events of 1997 and 1998. *Echinidinium delicatum* and SBD show increased fluxes during active upwelling intervals, particularly in the 'pulses' of 1998. *Spiniferites ramosus* and *Bitectatodinium spongium*, both species produced by autotrophic dinoflagellates, generally

respond positively to upwelling strength as well. Cysts type Cp are particularly abundant towards to end of the time series, during the weak upwelling event of 1999.



Some cyst taxa appear to be associated with secondary upwelling events only. *Echinidinium granulatum* and the cysts of cf. *E. carinata* show pronounced flux increases in Jul.-Aug. 1997, and fluxes of the cysts of *Protoperidinium fukuyoi* and *Gymnodinium nolleri* markedly increase in Jun.-Jul. 1998.

Another group of generally less abundant cyst taxa show higher fluxes at the onset of active upwelling intervals; these

include *Echinidinium aculeatum*, *Stelladinium robustum* and SBA (Fig. 5). Notably, increases of cysts of *Protoperidinium stellatum* fluxes preceding active upwelling intervals by up to ~1 month are recorded three times over the time series, although its maximum abundance occurs at the end of the strong upwelling event of 1997.

**4.4 Cyst production vs measured physico-chemical parameters (RDA$_{envi}$)**

The seasonality of dinoflagellate cyst production is analysed in relation to physico-chemical parameters in the RDA

(henceforth, 'RDA$_{envi}$') presented in Fig. 6. The ordination is significant as a whole (p = 0.007), with the first two RDA axes capturing 41.9 and 9.2% of the variance in species data (Fig. 6). The most important physico-chemical variables (significantly related to the dinoflagellate cyst data) are SST, bioSi, $PO_4$, and $CaCO_3$. The dominant gradient illustrated in the biplots is most easily described with regard to the state of upwelling. The lower left quadrant represents conditions of upwelling relaxation (with higher SST, lower nutrients and bioSi), whereas the upper right quadrant (lower SST, higher

nutrients and bioSi) corresponds to conditions of active upwelling. This interpretation is supported by the ordination of samples (Fig. 6B), which shows a clear distribution of samples along this gradient depending on the time of sediment collection.

The ordination of dinoflagellate cyst taxa along this gradient provides further evidence that *Brigantedinium* spp., *S. quanta*, SBC, *Q. concreta*, *Echinidinium* spp. and *Dubridinium* spp. are associated with conditions of active (strong) upwelling. Cyst

taxa that showed higher abundances in active upwelling intervals other than the strong upwelling of 1997, including secondary upwelling events, tend to be ordinated closer to the centroid. However, three cyst taxa have strong loadings along RDA$_{envi}$ axis 2: *E. granulatum* and the cyst of cf. *E. carinata* (with strong positive scores) and cyst type Cp (strong negative score). Even though RDA axis 2 is not statistically significant (Fig. 6C), the ordination suggests an ecological niche that may be different, that is, less related to the strength of upwelling, for these three taxa.

**4.5 Cyst production vs biological parameters (RDA$_{bio}$)**

Dinoflagellate cyst production was further investigated in the context of biological variables measured from water column and sediment trap sampling at the CARIACO station (Fig. 7). This RDA ('RDA$_{bio}$') was performed on square-root-transformed dinoflagellate cyst fluxes in order to give more weight to less abundant cyst taxa. RDA$_{bio}$ yields an ordination pattern of cyst taxa and significance levels similar to RDA$_{envi}$, with the first two axes capturing 25.6 and 9.2% of the variance

in the species data. Here too, both the first axis and the ordination as a whole are significant (p = 0.007), and the same underlying gradient of upwelling strength (bottom-left to top right: relaxed to active upwelling) is implied. Dinoflagellate cyst taxa previously found to be associated with active upwelling conditions (e.g., *Brigantedinium* spp., *S. quanta* and SBC)

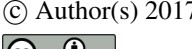



are ordinated close to indicators of diatom and zooplankton (mandibles and copepod eggs) abundance. Interestingly, the ordination along RDA axis 2 separates taxa such as *E. granulatum*, *S. mirabilis* and SBC on the positive side (positively correlated with prasinophytes), and the cysts of *G. nolleri*, *P. fukuyoi*, *D.* cf. *symmetrica* and cyst type Cp on the negative side, closer to chlorophytes, haptophytes and cryptophytes (Fig. 7).

## 5 Discussion

This study presents the first record of dinoflagellate cyst production and seasonality in the Caribbean Sea, and sheds new light on dinoflagellate ecology in the context of their physico-chemical and biological environment. The dinoflagellate cyst sedimentary record of the basin has previously been studied by Wall (1967), González et al. (2008) and Mertens et al. (2009a) at multi-centennial to millennial resolution. While many dinoflagellate cyst taxa identified in our sediment trap time series were previously reported from those studies, many others (total of 28, including types) are reported here for the first time (Table 1). Most of these 'new' taxa are rare, lack clear taxonomic characterization (types) or were simply described after the abovementioned studies were published. Conversely, a number of taxa previously reported in the basin (from sedimentary sequences that extend back to ~73 ka) were not encountered in the time series, namely *Achomosphaera* sp., *Ataxiodinium choane*, *Bitectatodinium tepikiense*, *Echinidinium euaxum*, *Echinidinium transparantum*, several *Impagidinium* species (*I. aculeatum*, *I. paradoxum*, *I. patulum*, *I. plicatum*, *I. sphaericum* and *I. strialatum*), *Islandinium minutum*, *Leipokatium invisitatum*, *Melitasphaeridium choanophorum*, *Nematosphaeropsis labyrinthus*, *Operculodinium israelianum*, *Selenopemphix selenoides*, some *Spiniferites* species (*S. elongatus*, *S. lazus*, *S. membranaceus*, *S. pachydermus* and *S. scabratus*), *Tectatodinium pellitum* and *Tuberculodinium vancampoae*.

The Cariaco Basin sediment trap record shows that heterotrophic cyst taxa, particularly of the family Protoperidiniaceae, dominate the dinoflagellate cyst assemblages, consistently with sediment trap studies from other upwelling systems (Zonneveld and Brummer, 2000; Susek et al., 2005; Ribeiro and Amorim, 2008; Pitcher and Joyce, 2009; Zonneveld et al., 2010; Bringué et al., 2013) or nutrient-rich regions without upwelling (e.g., Harland and Pudsey, 1999; Fujii and Matsuoka, 2006; Pospelova et al., 2010; Price and Pospelova, 2011). The Cariaco Basin is among the most productive systems in the world, with average cyst fluxes to the trap ($17.1 \times 10^3$ cysts m$^{-2}$ day$^{-1}$) most similar to fluxes reported from the NW Arabian Sea (Zonneveld and Brummer, 2000), about one order of magnitude higher than fluxes recorded at a station located ~ 370 km off Cape Blanc (Susek et al., 2005; Zonneveld et al., 2010) and about two orders of magnitude lower than a record from a near-shore station off Cape Columbine in the southern Benguela upwelling system (Pitcher and Joyce, 2009). When comparing records from the Cariaco Basin (this study) and the Santa Barbara Basin (Bringué et al. 2013), both sites characterized by seasonal upwelling, located at about the same distance from the nearest coast (~ 18 km) and using the same cyst extraction technique, cyst fluxes are about five times lower in the Cariaco Basin than in Santa Barbara Basin, where exceptionally high $C_{org}$ fluxes are almost twice as high as in the Cariaco Basin (Thunell et al., 2007).





### 5.1 Seasonality in dinoflagellate cyst production in the Cariaco Basin

### 5.1.1 The upwelling signal

The upwelling signal is by far the clearest signal observed in the dinoflagellate cyst record. Most dinoflagellate cyst species show a marked, positive response to upwelling strength (Figs. 3 and 5), an observation that is supported by multivariate

analyses (Fig. 6). Increased cyst fluxes during intervals of active upwelling are observed for taxa of both autotrophic and heterotrophic affinities, suggesting that nutrient supply is the most limiting factor for both groups, whether the effect of nutrient limitation is expressed directly (for autotrophs) or indirectly (for heterotrophs).

Biogenic silica, most of which is precipitated by planktonic organisms in the surface layers, is commonly used as a proxy for diatom abundance in marine systems, even though silicoflagellates and radiolaria also contribute to the bioSi flux (e,g.,

Nelson et al., 1995; Ragueneau et al. 2000; Romero et al. 2009). Diatoms have been shown to dominate primary production and the bioSi flux in the Cariaco Basin, particularly during upwelling events (e.g., Thunell et al., 2000, 2008; Romero et al. 2009; Taylor et al., 2012; Pinckney et al., 2015). In our record, dinoflagellate cyst fluxes of both autotrophic and heterotrophic affinities correlated positively and significantly with bioSi fluxes (Fig. 8A and B), although the relationship is much clearer for heterotrophs ($R^2 = 0.36$, $p = 0.000$), and even stronger when considering only *Brigantedinium* spp. fluxes

($R^2 = 0.47$, $p = 0.000$; Fig. 8C). This is consistent with previous observations from other coastal marine settings (e.g., Fujii and Matsuoka, 2006; Zonneveld et al., 2010; Price and Pospelova, 2011; Bringué et al., 2013; Heikkilä et al., 2014) and confirm that fluxes of cysts of heterotrophic affinity (*Brigantedinium* spp. in particular) can be used as indicators of primary (mostly diatom) productivity in the Cariaco Basin.

Diatom and silicoflagellate census data were also available for comparison with the cyst record, overlapping most of our

time series (Romero et al., 2009). Even though diatom abundances tend to increase during active upwelling intervals, fluxes of diatom frustules and silicoflagellate skeletons do not match the bioSi record, largely because of discrepancies observed during the weak upwelling event of 1999 (Fig. 3B). Several factors may explain this apparent mismatch. Firstly, the samples used to generate diatom and silicoflagellate data were extracted from Trap A (275 m, same as this study) until Nov. 1997, but from a deeper trap (Trap B, 455 m) afterwards (Fig. 3B). However, it is unlikely that a difference in sedimentation

between traps (e.g., by lateral advection of sediment at depth, or differences in trapping efficiency) played a role here since bioSi fluxes were also low in the Trap B record (Romero et al., 2009). Radiolaria fluxes were not quantified, and even modest numbers of those typically larger (> 150 μm) tests may also account for some discrepancies between the records. More importantly, the large increases in diatom fluxes recorded in 1999 are mainly attributable to small and/or weakly silicified species, namely *Cyclotella litoralis* and resting spores of *Chaetoceros* (Romero et al., 2009), which may not result

in elevated mass fluxes of bioSi.

The fact that fluxes of heterotrophic dinoflagellate cysts are strongly, positively correlated with bioSi but not with fluxes of diatom frustules suggests that heterotrophic dinoflagellates respond more to total diatom biomass than to the availability of any particular diatom species in the Cariaco Basin. It is also possible that particles ballasted with bioSi, together with



CaCO₃-rich particles (also a significant variable in RDA$_{envi}$; Fig. 6) facilitated the export of dinoflagellate cysts to the depths (the 'Ballast mineral' hypothesis; e.g., Thunell et al., 2007).

### 5.1.2 Indicators of upwelling

While nearly all dinoflagellate cyst taxa respond positively to upwelling in the system, the detailed response of each species

vary depending on upwelling strength. In addition to *Brigantedinium* spp., several cyst taxa from the family Protoperidiniaceae are clearly associated with active upwelling (with the highest fluxes recorded during moderate to strong upwelling events), including *S. quanta*, *Q. concreta* and *S. nephroides*, which are known to have a particularly good relationship with diatoms (e.g., Jacobson and Anderson, 1986; Dale et al., 2002; Radi and de Vernal, 2008; Pospelova et al., 2008, 2010; Limoges et al., 2010; Price and Pospelova, 2011; Bringué et al., 2013, 2014). Our SBC type shows a clear

preference for moderate to strong upwelling conditions as well (Fig. 5).

Two members of the subfamily Diplopsalioideae (*Dubridinium* spp. and cyst of cf. *Diplopelta* sp.) also seem to favor active upwelling conditions, although they tend to appear rather sporadically in the record and not only during active upwelling intervals (Fig. 5). *Dubridinium caperatum*, the most abundant *Dubridinium* species in our record, is usually found in highest abundances in the vicinity of active upwelling cells (e.g., Zonneveld et al., 2013). Other *Dubridinium* species tend to be

abundant in coastal areas where food sources are diverse (e.g., Naustvoll, 2000), including in eutrophic and heavily polluted embayments (e.g., Matsuoka, 1999; Pospelova et al., 2005; Dale, 2009; Pospelova and Kim, 2010; Krepakevich and Pospelova, 2010). Our record suggests that these two diplopsalid taxa reflect active upwelling conditions in the Cariaco Basin.

Several 'spiny brown' cysts (*E. aculeatum*, *E. delicatum*, *E. granulatum*, *Echinidinium* spp. and SBD) also show a strong

association with active upwelling conditions, but they seem to respond more strongly to weaker upwelling conditions, with the highest fluxes recorded either at the onset or during weak or secondary upwelling events (Fig. 5). Many *Echinidinium* species are commonly found in upwelling regions worldwide (e.g., Zonneveld et al., 2013) and in a sediment trap study from the California Current upwelling system, fluxes of *E. aculeatum*, *E. delicatum* and *Echinidinium* spp. were highest when upwelling was active but not at its strongest (Bringué et al., 2013). *Echinidinium granulatum* was observed during active

upwelling in the Arabian Sea (Zonneveld and Brummer, 2000), but no clear link was observed off Cape Blanc, where it remained nonetheless associated with elevated nitrate and C$_{org}$ concentrations (Zonneveld et al., 2010). Therefore, in the Cariaco Basin, these spiny brown taxa may be good indicators of weak to moderate upwelling conditions.

Three taxa of autotrophic affinity (*B. spongium*, *S. ramosus* and cyst of cf. *E. carinata*) also show higher abundances during active upwelling intervals (Fig. 5.), likely as a direct response to nutrient loading. A common species in our record, *B.*

*spongium* is known to thrive in warm (year-round SST > 20°C), nutrient-rich environments (e.g., Zonneveld and Jurkschat, 1999; Zonneveld et al., 2013) and was also documented to produce cysts during times of active upwelling in sediment trap studies from the Somali Basin (NW Arabian Sea; Zonneveld and Brummer, 2000) and off Cape Blanc (NW Africa; Susek et al., 2005; Zonneveld et al., 2010). The detailed response of *S. ramosus* and cysts of cf. *E. carinata* to upwelling is less clear.




*Spiniferites ramosus* is a cosmopolitan species (Zonneveld et al., 2013) with complex seasonal patterns that vary by site (Pospelova et al., in press). In our record, *S. ramosus* shows no clear relationship to SST (Fig. 6) but responds to nutrient input, with modest increases in fluxes of cysts with cell content (i.e., recently produced and not yet excysted) recorded during almost all upwelling events, regardless of upwelling intensity or duration (Fig. 5). The single, large bloom of cf. *E.*

*carinata* observed during the secondary upwelling event of 1997 (Fig. 5) occurred at a time when nutrients were available below a shallow MLD (< 20 m), during an upwelling event that yielded very little Chl *a* levels (Fig. 2). It has been suggested that environmental affinities of *E. carinata* are similar to those of *Scrippsiella trochoidea* with optimal SST range of 18-21°C (Shin et al., 2012), which would correspond to temperatures observed below 50 m during the Jul.-Aug. 1997 bloom. This may suggest a deeper habitat for cf. *E. carinata* in the Cariaco Basin, rather than a direct association with upwelling.

**5.2 Interactions between dinoflagellates and other major planktonic groups**

Marine food webs are complex trophic systems in which dinoflagellates interact with a variety of organisms, including components of the picoplankton, phytoplankton, and micro- to macrozooplankton. Autotrophic dinoflagellates, as primary producers, compete for resources (nutrients) with other phytoplankton groups (diatoms, haptophytes, chlorophytes and so forth). Diatoms typically dominate primary production in upwelling systems, including in the Cariaco Basin (e.g., Romero et

al., 2009; Pinckney et al., 2015), while autotrophic dinoflagellates and haptophytes are generally thought to be more abundant during, or at the transition to, upwelling relaxation (e.g., Goñi et al., 2003; Smayda and Trainer, 2010; Bringué et al., 2013). In a study of the Si cycle and Si:C:N ratios of sinking particles in the Cariaco Basin, Thunell et al. (2008) demonstrated that silicic acid (a macronutrient essential for diatom growth but not used by dinoflagellates) is the limiting nutrient in the upper 50 m of the water column during active upwelling, whereas nitrates limit primary production in

summer/fall. In our record, autotrophic dinoflagellates (cysts and motiles alike), do not appear to compete with diatom for resources, as both groups respond positively to upwelling (Figs. 3 & 5), and fluxes of autotrophic cysts and bioSi are positively correlated (Fig. 8). This is consistent with silicic acid limiting diatom production during active upwelling, and nitrate limitation affecting both diatoms and autotrophic dinoflagellates during upwelling relaxation.

Heterotrophic (and mixotrophic) dinoflagellates exert a significant, if not dominant, grazing impact in upwelling and other

productive ecosystems (Jeong, 1999; Sherr and Sherr, 2007; Calbet, 2008). Besides their role as grazers on phytoplankton, they can also prey on copepod eggs and early naupliar stages, and serve as prey for meso- to macrozooplankton or for other dinoflagellates (e.g., Jeong, 1999). In a sediment trap study off Cape Blanc, Zonneveld et al. (2010) investigated the potential food sources of heterotrophic dinoflagellates using diatom census data and measurements of bioSi, $C_{org}$ and $CaCO_3$. They observed a significant relationship between cyst and total diatom valves accumulation rates, including for

*Brigantedinium* spp. and *S. quanta*, as well as several protoperidinioid cyst taxa possibly associated with haptophytes (suggested by a correlation with $CaCO_3$ fluxes), including *P. stellatum*. In this study, potential interactions between heterotrophic dinoflagellates and other important planktonic groups are investigated using $RDA_{bio}$ (Fig. 7) and by comparing cyst fluxes with photopigment data of Pinckney et al. (2015). As discussed above, most of the abundant heterotrophic cyst



taxa are directly associated with active upwelling conditions, when diatom dominates primary production. Not surprisingly, indicators of zooplankton abundance (mandibles and copepod eggs) are ordinated close to the 'active upwelling pole' of the implied gradient in the RDA$_{bio}$ biplot (upper right quadrant in Fig. 7), together with diatoms. However, this only suggests co-occurrence and we cannot comment further on causal (trophic) relationships.

Several other heterotrophic cyst taxa in the record show increased abundances that do not appear to be directly related to diatom availability. Cysts of *P. stellatum* show repeated increases in abundance during upwelling relaxation, generally within ~ one month of upwelling onset (Fig. 5). Interestingly, two of the pronounced increases in *P. stellatum* cysts occur at times when haptophytes (including coccolithophorids) where recorded in relatively large quantities in the water column (Pinckney et al., 2015). In a similar manner than in Zonneveld et al. (2010), *P. stellatum* is ordinated close to CaCO$_3$ is in

RDA$_{envi}$ (Fig. 6), even though CaCO$_3$ is not clearly related to haptophyte productivity in the basin (Goñi et al., 2009). Despite the absence of increase in *P. stellatum* cyst flux during the May-Jul. 1998 subsurface bloom of coccolithophes (Pinckney et al., 2015), there is growing evidence that haptophytes may serve as a food source for *P. stellatum* (Zonneveld et al., 2010).

The only noticeable increases in fluxes of cysts of *P. fukuyoi* and *G. nolleri* took place during the short, secondary upwelling

event of Jul. 1998 (Fig. 5). These increases follow a peak in ciliate fluxes (Fig. 3) and occurred at a time when coccolithophores, cryptophytes and chlorophytes were abundant below the MLD; prasinophytes were also present but in lower abundances than during most of the record (Pinckney et al., 2015). In RDA$_{bio}$, the cysts of *P. fukuyoi* and *G. nolleri* are ordinated in the same quadrant (lower left) as haptophytes, cryptophytes and chlorophytes (Fig. 7). Although these organisms were also abundant at other times in our record, their co-occurrence suggests possible trophic relationships, and

the fact that all those potential food sources were present only below the MLD may suggest that *P. fukuyoi* and *G. nolleri* may dwell in, or access, deeper waters. Interestingly, higher fluxes of cysts of *P. fukuyoi* were also recorded in the Santa Barbara Basin under similar circumstances, when a short, secondary upwelling pulse interrupted relaxed upwelling conditions (Bringué et al., 2013). Other sediment trap records of *P. fukuyoi* cyst production from the Strait of Georgia (Pospelova et al., 2010) and Saanich Inlet (Price and Pospelova, 2011) in the productive waters of coastal southwestern

Canada also indicate higher fluxes during the warmest months of the year (Mertens et al., 2013).

*Echinidinium granulatum* also showed a single interval with marked increased abundances, during the secondary upwelling event of Jul.-Aug. 1997 (Fig. 5). Of the available data, the only planktonic groups that were noticeably abundant at this time were diatoms (mostly *C. litoralis*; Romero et al., 2009) and prasinophytes, which may act as food sources for *E. granulatum*. Cysts type Cp (possibly the cysts of *Cochlodinium polykrikoides* sensu Li et al. 2015) have a strong, negative loading along

RDA$_{bio}$ axis 2 with no obvious link with other planktonic groups (Fig. 7). This taxon was observed almost exclusively at the end of the time series, during the weak upwelling event of 1999 which resulted in modestly elevated Chl *a* concentrations (Fig. 2) and large fluxes of (mostly small or weakly silicified) diatom frustules and silicoflagellates that were, however, not reflected in the bioSi record (Fig. 3). One of the major species responsible for harmful algal blooms, *C. polykrikoides* exhibits a wide range of temperatures over which it can survive, and SST conditions in the Cariaco Basin fall within its



optimal range (e.g., Matsuoka and Iwataki, 2004; Kudela and Gobler, 2012). Increases in cysts type Cp fluxes during that event is similar in timing and magnitude to that of *Brigantedinium* spp., the dominant cyst taxa in our record, but the exact factor(s) behind such a large increase in cyst type Cp abundances remains unclear.

**5.3 Effect of NAO and ENSO on dinoflagellates**

The NOA and ENSO phenomena have been documented to modulate precipitation in the Caribbean, and El Niño events may influence the basin's hydrography with a one year lag (e.g., Enfield and Mayer, 1997; Giannini et al., 2001; Astor et al., 2013). Summarizing observations at the CARIACO station over 14 years, Taylor et al. (2012) found only modest correlations between, on one hand, NAO and ENSO (with one year lag) indices, and hydrographical and biological measurements in the basin. Romero et al. (2009) reported a major change in the composition of diatom assemblages coeval

with the 1997/98 ENSO event, with lower diatom production compared to the previous year and an increase in the relative contribution of pelagic species more commonly found in Caribbean surface waters.

Over the duration of our time series, direct comparison (i.e., with no time lag applied) of NAO index and Multivariate ENSO Index (MEI) to SST measurements yielded significant but weak, negative correlations (Pearson's $r$ = -0.33, p = 0.011 for NOA; $r$ = -0.28, p = 0.031 for MEI). Visual examination of variations in dinoflagellate cyst production does not reveal any

apparent link with NAO and ENSO (Fig. 9). Numerically, total dinoflagellate cyst fluxes and total heterotrophic cyst fluxes are weakly, positively correlated with the NAO index ($r$ = 0.33, p = 0.010 and $r$ = 0.34, p = 0.008, respectively), while total autotrophic cyst fluxes show a weak, positive correlation with MEI ($r$ = 0.40, p = 0.002). Thus, there is little evidence for any 'direct' (with no time lag) influence of NAO or ENSO on dinoflagellate cyst production in the basin.

However, our record suggests that the 1997/98 El Niño event was rather expressed with a one year lag in the basin, with total

cyst fluxes of the year 1999 being the lowest recorded between 1996 and 2009 (Fig. 9; Bringué, Thunell and Pospelova, unpublished data). A one year delay in ENSO effects has been observed for precipitation and SST patterns in the Caribbean (with dryer summers on the year of an El Niño event and wetter conditions the following summer; Enfield and Mayer, 1997; Giannini et al., 2001; Fig. 9) and consistent with observations at the CARIACO station (e.g., Müller-Karger, 2004; Taylor et al., 2012). A similar 'damping' effect was observed in the Santa Barbara Basin, where the strong 1997/98 El Niño event

hampered upwelling and resulted in greatly reduced primary production (e.g., Lange et al., 2000; Shipe et al., 2002). The dinoflagellate cyst response to the 1997/98 El Niño event was also documented in the Strait of Georgia, (BC, Canada) by Pospelova et al. (2010). At this location, positive SST anomalies in the region and low snow accumulation over the Frasier River watershed associated with the 1997/98 El Niño winter (e.g., Foreman et al., 2001) resulted in higher SST and SSS conditions in the Strait in the spring of 1998, and fluxes of cysts produced by some autotrophic taxa (including

*Spiniferites* spp. and the cysts of *P. dalei*) and several heterotrophic taxa (*E. delicatum*, *Echinidinium* spp., *S. nephroides*, *S. quanta* and the cysts of *P. fukuyoi* – as 'cyst type A'), benefitting from an early start of their production season, were higher than in 1996 and 1997 (Pospelova et al., 2010). Apart from the fact that the time lag between ENSO events and their influence on the Cariaco Basin may be longer (about a year; e.g., Enfield and Mayer, 1997; Giannini et al., 2001) than on the





Strait of Georgia (likely between ~ 4 and 10 months; e.g., Bograd and Lynn, 2001; Royer, 2005), the contrasting environmental settings between the two sites likely explains the very different response in dinoflagellate cyst production to the strong 1997/98 El Niño event. While cyst production of many dinoflagellates appear to have been promoted in the Strait of Georgia by a longer production season, weaker upwelling in the Cariaco Basin in 1998 and 1999 (compared to 1997)

resulted in lower total cyst fluxes (Fig. 9). However, the unusually high production of cysts type Cp in Feb.-May 1999 may be related to delayed effects of this strong 1997/98 El Niño event, although the exact mechanism remains unknown. Clearly, longer time series are needed to properly investigate ENSO effects in the basin and interannual variability in general.

## 6 Conclusions

This study documents dinoflagellate cyst production in the Cariaco Basin over a period of ~2.5 years. The sediment trap

record shows that the basin harbors very diverse and abundant communities of cyst-producing dinoflagellates, with over 60 morphotypes identified (including ~30 reported in the basin for the first time) and cyst fluxes averaging $17.1 \times 10^3$ cysts m$^{-2}$ day$^{-1}$, and up to $69.2 \times 10^3$ cysts m$^{-2}$ day$^{-1}$ under active upwelling conditions.

In the basin, dinoflagellate cyst associations are dominated by cysts of heterotrophic affinity, particularly *Brigantedinium* spp. which alone account for over half the total trap assemblage. The seasonality in dinoflagellate cyst production strongly

reflects variations in upwelling strength, with increases in cyst fluxes observed during active upwelling intervals. Fluxes of heterotrophic dinoflagellate cysts show a strong, positive correlation with biogenic silica but not with fluxes of diatom frustules, indicating that heterotrophic dinoflagellates respond more to total diatom biomass than to the availability of any particular species.

Dinoflagellate cyst taxa clearly associated with active upwelling conditions include *Brigantedinium* spp., *Selenopemphix*

*quanta*, *Selenopemphix nephroides*, *Quinquecuspis concreta* and *Dubridinium* spp. Cyst taxa of autotrophic affinity also respond positively to nutrient loading during active upwelling, with *Bitectatodinium spongium* showing the clearest response. Several 'spiny brown' cysts, including *E. aculeatum*, *E. delicatum*, *E. granulatum* and *Echinidinium* spp. show a closer affinity to weaker upwelling conditions.

New insights on possible trophic interactions are provided by comparing dinoflagellate cyst production with abundances of

other major components of the planktonic food web at the CARIACO station. Autotrophic dinoflagellates do not appear to compete for resources (nutrients) with diatoms in the basin, as silicic acid limits diatom production during active upwelling intervals. Zooplankton abundance indicators (mandibles and copepod eggs) co-occur with cyst taxa associated with active upwelling conditions (mostly Protoperidiniaceae), when diatom productivity is high. In particular, the production of *Protoperidinium stellatum* cysts may be associated with haptophytes and/or $CaCO_3$ fluxes. The bulk of *Protoperidinium*

*fukuyoi* and *Gymnodinium nolleri* cyst production took place during a short, secondary upwelling event that was marked by high abundances of coccolithophores, cryptophytes and chlorophytes, suggesting that these groups may serve as food sources for *P. fukuyoi* and *G. nolleri*.

On interannual time scales, dinoflagellate cyst production seems to be influenced by the strong 1997/98 El Niño event, with a one year lag. While seasonal variations associated with upwelling constitutes the dominant signal in the cyst record, interannual variability is also considerable and longer time series are needed in order to fully understand the environmental controls on dinoflagellate cyst production. Nevertheless, this work expands our knowledge of cyst-producing dinoflagellate ecology, helping the interpretation of fossil assemblages from the basin's sedimentary record and worldwide.

## Acknowledgements

This work was funded by a Natural Sciences and Engineering Research Council of Canada (NSERC) Postdoctoral Fellowship (PDF) and an Advanced Support Program for Innovative Research Excellence (ASPIRE I – Track IIb) from the U. of South Carolina to MB, and an NSERC Discovery grant to VP. The CARIACO sediment trapping program was supported by National Science Foundation (NSF; grants OCE 9401537, OCE 9729697 and OCE 1258991). The authors wish to thank Claudia Benitez-Nelson (U. of South Carolina) for providing mixed layer depth data and for helpful discussions on the hydrology of the basin. Kenneth Mertens (IFREMER – Concarneau) is kindly thanked for his help in dinoflagellate cyst identification. The crew of the R/V Hermano Gines is also gratefully acknowledged for all operations at sea.

## Supplementary material

Supplementary material provided with this article includes total dinoflagellate cyst fluxes and cyst counts (Table S1), as well as all physico-chemical and biological variables used in analyses (Table S2).

## Author contribution

MB, RCT and VP designed the project. RCT and ET organized and performed sediment trap deployments, and generated geochemical data on trap samples. VP provided lab (equipment and consumables) for the palynological treatment of samples. JLP and OER provided photopigment and siliceous microfossil data, respectively. MB selected, split, processed and counted all samples, analysed the data, generated all figures and wrote the manuscript. VP, RCT, OER and JLP reviewed the manuscript.

## Competing interests

The authors declare that they have no conflict of interest.




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



**Table 1.** List of dinoflagellate cyst taxa identified in the Cariaco sediment trap time series, and their motile equivalents (e.g., Zonneveld and Pospelova, 2015).

| Dinoflagellate cyst (Paleontological name) | Motile dinoflagellate (Biological name) |
|---|---|
| - | *Archaeperidinium constrictum* [a],* |
| - | cf. *Biecheleria* sp. [a,b] |
| *Bitectatodinium spongium* | Gonyaulacoid group |
| *Brigantedinium cariacoense* | *Protoperidinium avellanum* |
| *Brigantedinium irregulare* [a] | *Protoperidinium denticulatum* |
| *Brigantedinium majusculum* [a] | *Protoperidinium* sp. indet. |
| *Brigantedinium simplex* | *Protoperdinium conicoides* |
| *Brigantedinium* spp. | ?*Protoperidinium* spp. |
| - | cf. *Diplopelta* sp. [a] |
| - | *Diplopelta* cf. *symmetrica* [a] |
| *Dubridinium caperatum* [a] | *Preperidinium meunieri* |
| *Dubridinium* cf. *ulsterum* [a] | Diplopsalid group |
| *Dubridinium* spp. [a],* | Diplopsalid group |
| *Echinidinium aculeatum* | Diplopsalid or Protoperidinioid group |
| *Echinidinium delicatum* | Diplopsalid or Protoperidinioid group |
| *Echinidinium granulatum* | Diplopsalid or Protoperidinioid group |
| *Echinidinium zonneveldiae* [a] | Diplopsalid or Protoperidinioid group |
| *Echinidinium* cf. *delicatum* [a] | Diplopsalid or Protoperidinioid group |
| *Echinidinium* spp. * | Diplopsalid or Protoperidinioid group |
| - | cf. *Ensiculifera carinata* |
| - | *Gymnodinium nolleri* |
| *Lejeunecysta marieae* * | Protoperidinioid group |
| *Lejeunecysta oliva* * | Protoperidinioid group |
| ?*Lejeunecysta* sp. | Protoperidinioid group |
| *Lingulodinium machaerophorum* * | *Lingulodinium polyedrum* |
| *Operculodinium centrocarpum* sensu Wall & Dale 1966 * | *Protoceratium reticulatum* |
| *Operculodinium centrocarpum* reduced processes * | *Protoceratium reticulatum* |
| - | *Pentapharsodinium dalei* * |
| - | *Polykrikos kofoidii* sensu Matsuoka et al. 2009 [a],* |
| *Polysphaeridium zoharyi* * | *Pyrodinium bahamense* var. *bahamense* |
| - | *Protoperidinium americanum* [a],* |
| - | *Protoperidinium fukuyoi* [a] |
| - | *Protoperidinium nudum* [a] |
| - | *Protoperidinium oblongum* (cordate cyst type sensu Wall & Dale 1968) * |
| - | *Protoperidinium oblongum* (rhombic cyst type sensu Wall & Dale 1968) * |
| - | *Protoperidinium stellatum* |
| - | *Protoperidinium thulense* [a] |
| - | *Protoperidinium* sp. C [a],* |
| *Quinquecuspis concreta* * | *Protoperidinium leonis* |
| *Selenopemphix nephroides* | *Protoperidinium subinerme* |
| *Selenopemphix quanta* | *Protoperidinium conicum* |
| *Selenopemphix undulata* [a] | *Protoperidinium* sp. indet. |
| *Spiniferites belerius* [a] | *Gonyaulax scrippsiae* |
| *Spiniferites bentorii* | *Gonyaulax spinifera* complex |
| *Spiniferites bulloideus* | *Gonyaulax spinifera* complex |
| *Spiniferites hyperacanthus* | *Gonyaulax spinifera* complex |
| *Spiniferites mirabilis* | *Gonyaulax spinifera* complex |
| *Spiniferites ramosus* | *Gonyaulax spinifera* complex |
| *Spiniferites* spp. * | *Gonyaulax* spp. |
| *Stelladinium robustum* [a] | *Protoperidinium* sp. indet. |
| *Votadinium calvum* * | *Protoperidinium oblongum* |
| *Votadinium spinosum* [a],* | *Protoperidinium claudicans* |
| Cyst type Cp [a] | (*Cochlodinium polykrikoides*?) |
| Cyst type M1 [a],* | *Protoperidinium* sp. indet. |
| Spiny brown type A [a] | (*Archaeperidinium bailongense*?) |
| Spiny brown type C [a] | ?*Protoperidinium* sp. indet. |
| Spiny brown type D [a] | ?*Protoperidinium* sp. indet. |
| Spiny browns (other) * | ?Protoperidinoid group |
| Protoperidinoids * | ?*Protoperidinium* spp. indet. |
| Unknown cysts (heterotrophs) * | - |
| Unknown cysts (autotrophs) * | - |
| Organic linings of calcareous dinoflagellate [b] | - |

[a] Taxa reported in the Cariaco Basin for the first time
[b] Taxa not included in cyst counts
* Taxa never exceeding 1.5% of cyst assemblages




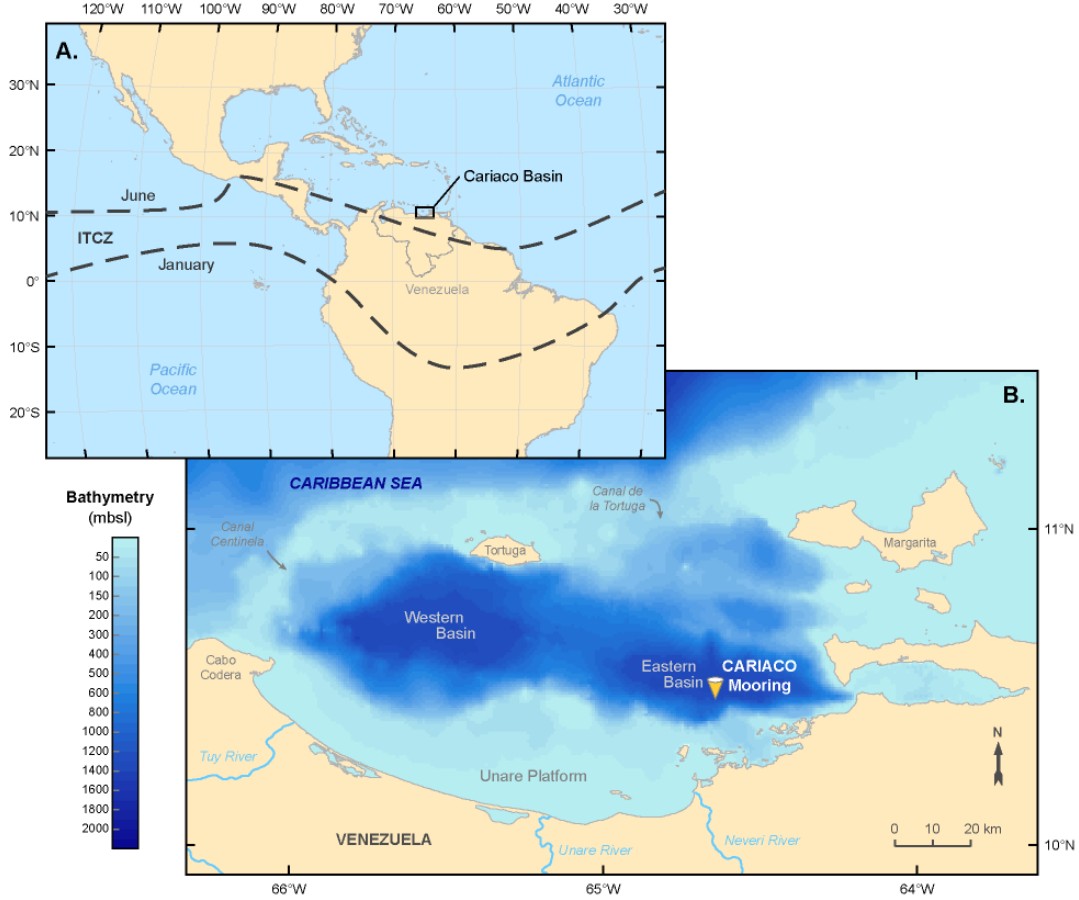

**Fig. 1. A.** Location of the Cariaco Basin in the southern Caribbean Sea, with boreal winter and summer positions of the Intertropical Convergence Zone (ITCZ) indicated as dashed lines. **B.** Map showing the main bathymetric features, local rivers emptying in the basin and the location of CARIACO station in the eastern sub-basin.





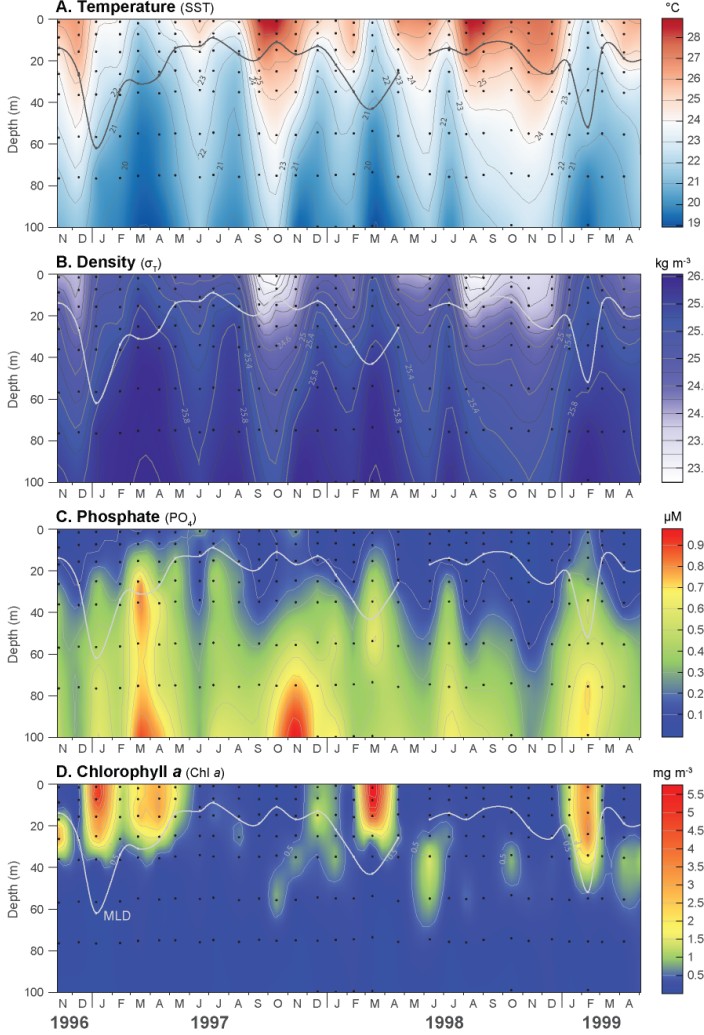

**Fig. 2**. Contour plots of SST (A), $\sigma_T$ (B), $PO_4$ (C) and Chl *a* (D) over the upper 100 m from monthly water column sampling at the CARIACO station. The grey (in A) or white (B–D) line indicates the mixed layer depth (MLD).

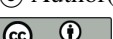

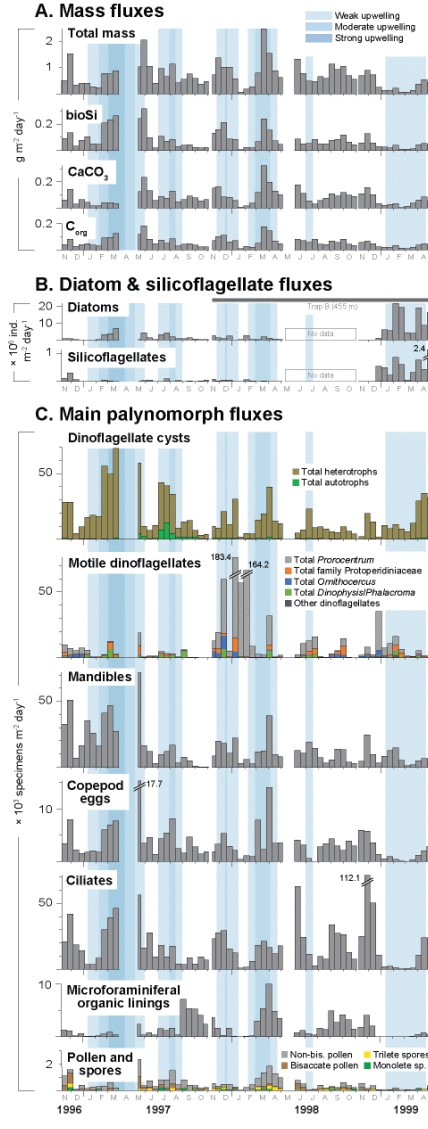

**Fig. 3.** Main fluxes to the trap. **A.** Fluxes of total sediment mass, biogenic silica, calcium carbonate and organic carbon. **B.** Diatom and silicoflagellate fluxes (from Romero et al., 2009). **C.** Fluxes of the dominant groups of palynomorphs. Vertical, blue shaded areas highlight intervals of active upwelling. Breaks in the horizontal axes indicate gaps in the time series.





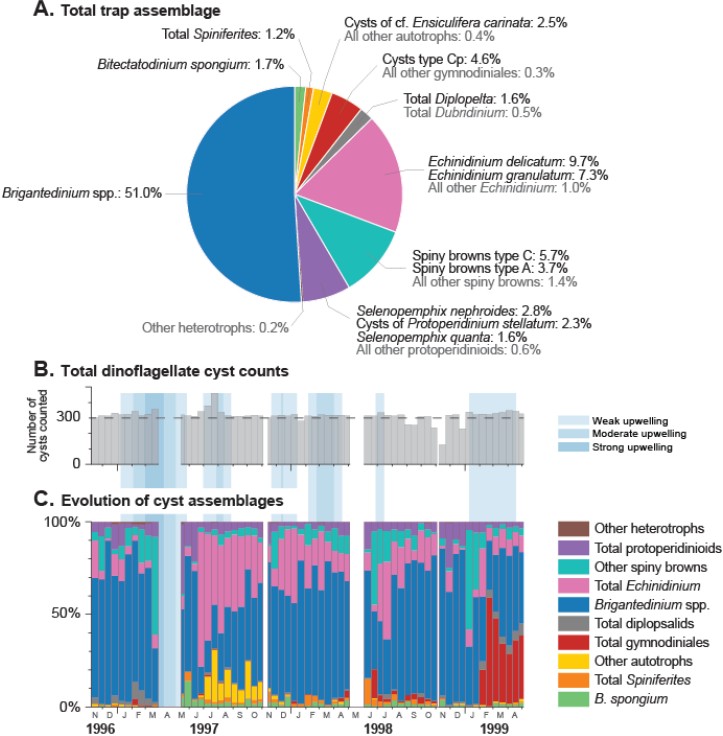

**Fig. 4.** Dinoflagellate cyst assemblages in the Cariaco Basin over the ~ 2.5 years of the sediment trap time series. **A.** Total trap assemblage. **B.** Cyst counts per sample, indicating the few samples < 300 which should be interpreted with caution. **C.** Contributions from the main groups of dinoflagellate cysts. Blue shaded areas indicate intervals of active upwelling, as in Fig. 3.





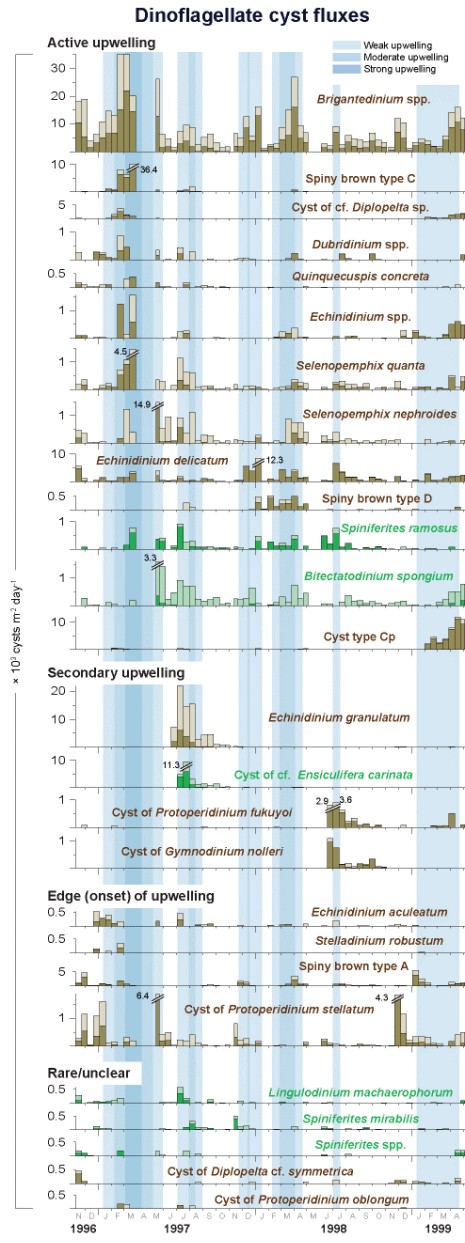

**Fig. 5.** Fluxes of selected dinoflagellate cyst taxa in the sediment trap samples. Cyst taxa are organized in groups (active upwelling, secondary upwelling, edge (onset) of upwelling and rare/unclear) based on their abundance patterns relative to upwelling intervals (indicated as blue, shaded areas). Cyst taxa of autotrophic (heterotrophic) affinity are shown in green (brown), with darker shades indicating cysts with cell content.



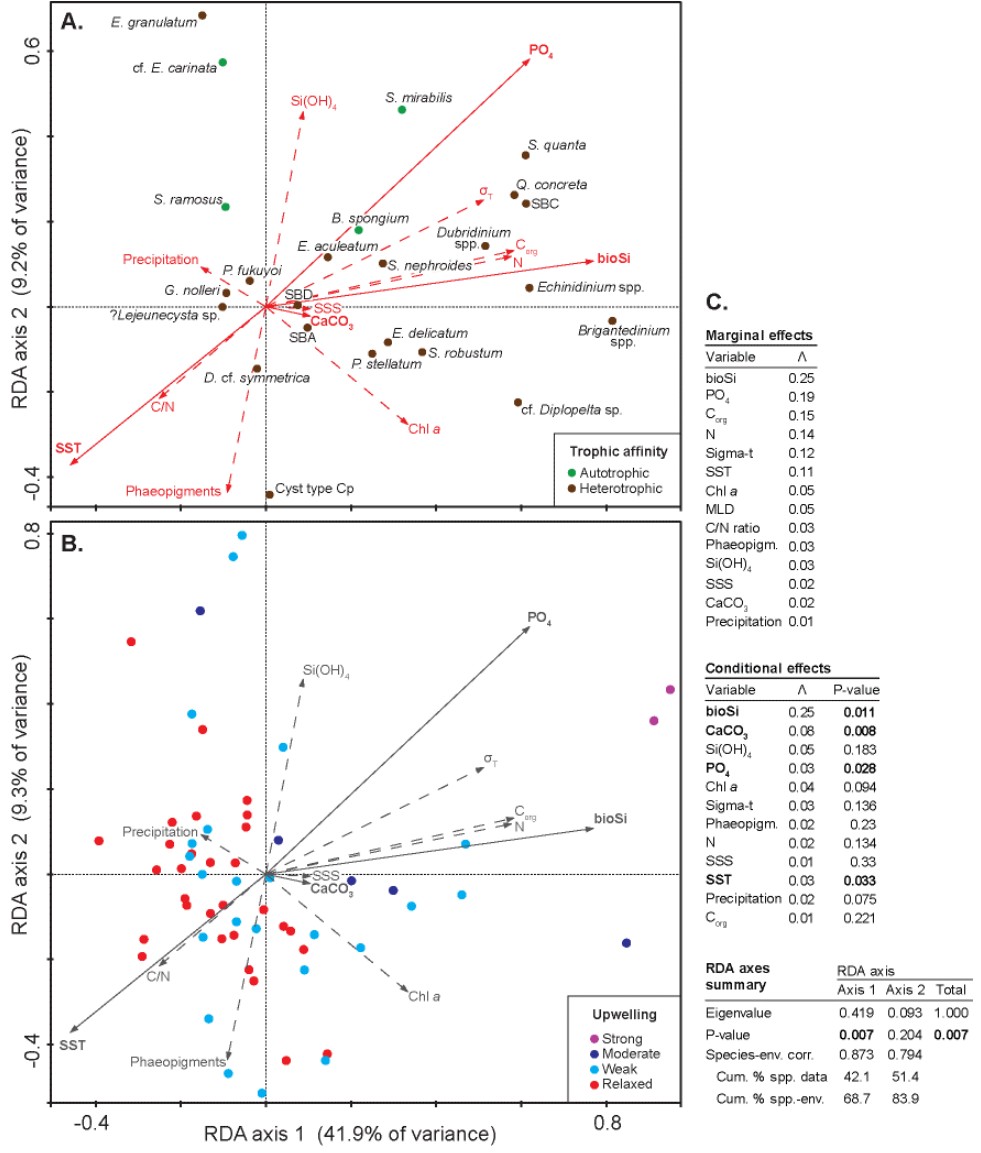

**Fig. 6.** Redundancy Analysis (RDA) performed on dinoflagellate cyst fluxes, using physico-chemical parameters as environmental variables (RDA_envi). **A.** Ordination of dinoflagellate cyst taxa and environmental variables. **B.** Ordination of samples. **C.** Marginal effects, conditional effects and summary of axes statistics. Statistically significant variables and axes are shown as solid arrows and bold labels.



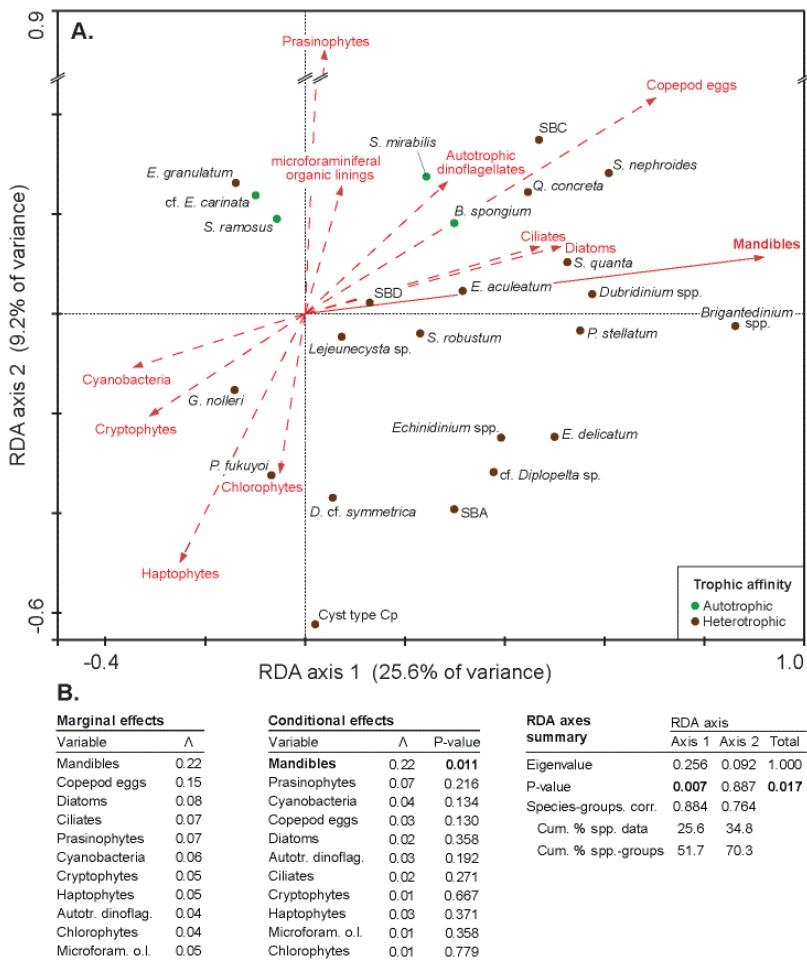

**Fig. 7.** RDA of square-root-transformed dinoflagellate cyst fluxes, in the context of other major groups of the planktonic food web (RDA$_{bio}$). **A.** Ordination of cyst taxa and biological variables, obtained from palynological data (this study) and photopigment data (including 'autotrophic dinoflagellates' and 'diatoms', from Pinckney et al., 2015). **B.** Marginal effects, conditional effects and summary of RDA axes statistics. 'Mandibles', the only statistically significant variable, is shown as a solid arrows with bold label.



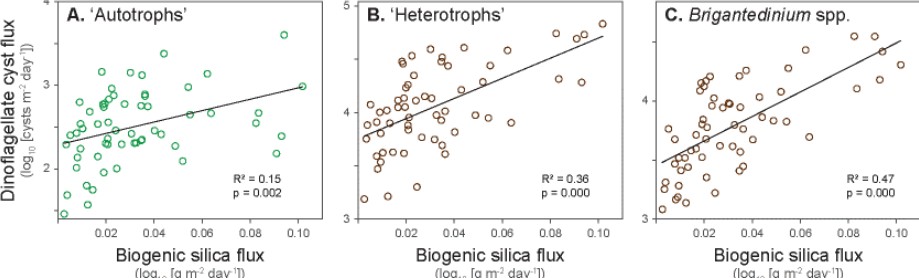

**Fig. 8.** Relationships between biogenic silica flux and fluxes of autotrophic taxa (**A**), heterotrophic taxa (**B**) and *Brigantedinium* spp. (**C**). All fluxes are log-transformed and coefficients of determination ($R^2$) are calculated after linear regressions.

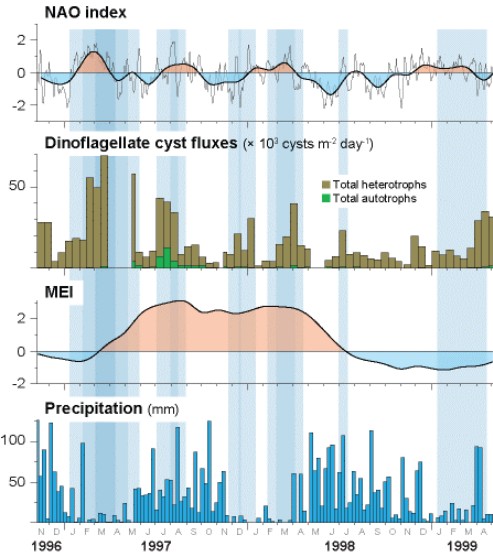

**Fig. 9.** Comparison of dinoflagellate cyst fluxes to the trap with the North Atlantic Oscillation (NAO) index, the Multivariate ENSO Index (MEI) and weekly precipitation over the Cariaco Basin. NAO index is shown as daily values (thin grey line) and monthly averages (thick, 10 black line), while MEI is drawn from monthly values. Blue shaded areas highlight active upwelling intervals, as in previous figures.