# Peer review of "Physico-chemical and biological factors influencing dinoflagellate cyst production in the Cariaco Basin"

_Biogeosciences, 2017_

## Referee Comment (RC1) · B. Dale (Referee) · 14 Dec 2017

Dinoflagellate cysts are increasingly used for marine paleoenvironmental interpretations, based on "cyst signals" recognized from studies of cyst distributions in the present-day ocean. These include signals for dissolved nutrients, SST, salinity, and distance from shore, largely developed by comparing cyst assemblages in surface sediments with environmental conditions in correspondingly overlying surface waters. The accuracy of signals inferred from such comparisons is dependent on the level of our understanding of which cysts are formed where and when in the ecological system. There are obvious limitations to simply comparing cysts in bottom sediments

with overlying surface water: cysts may be transported significant distances from their point of origin in the plankton prior to being deposited in the sediment (especially in deep oceanic or otherwise turbulent waters), possibly obscuring evidence of where they were formed; the age of cysts in bottom sediments is dependent on rates of sedimentation, potentially obscuring when the cysts were formed. The hydrographic data available for comparison with the sediments are of varied quality. Sediment traps as deployed here can provide more plausible information, while also recognizing certain limitations to this approach, too, as the authors do. By choosing the upper trap of this array (at 275m), should reduce the possibility for long-transport of the cysts sampled, and the succession of cup- collections is expected to show the approximate time of cyst formations in the upper water mass. The series of hydrographic data collected at the same station and times should allow close cyst/environment comparisons. The results produced seem to confirm a by now well established nutrient signal with heterotrophic species dominating the cyst signal with increased nutrients from upwelling – with some interesting first suggestions of nuances involving different species with differing degrees of upwelling. This is excellent work that pushes our understanding a step further by suggesting when and where the reported cysts may have formed, allowing this to be correlated to the corresponding environmental parameters, in at least this one location. Furthermore, including data for other plankton groups has allowed some suggestions of possible links to the very complex marine food web. This work sets the bar high for the many more such studies that will be needed to increase our understanding of just how precise the cyst signals may be. As a reviewer, I could wish that more studies were as well designed, carried out and reported as this, and the authors are to be congratulated. The only possible improvements I could suggest for the paper would be to indicate any currents affecting the water entering the basin (possible cyst transport?), and to indicate if possible what proportion of the cysts trapped had "fresh" cell contents. Our experience with sediment traps is that cell contents may be good indicators of "newly-formed" cysts. The authors perhaps wisely avoid serious attempts to relate this evidence from one trap depth to the bottom sediments that form

the broader reason for carrying out such studies, but I hope they will continue to investigate the lower trap samples eventually to see how accurately the cyst signal they identify is translated into the sediment.

---

## Referee Comment (RC2) · Dr. Ribeiro (Referee) · 11 Mar 2018

General comments

Dinoflagellates cysts are ubiquitous in coastal ecosystems worldwide, and have been increasingly used in palaeoenvironmental studies as indicators of past changes in e.g. sea surface temperature, salinity, nutrient status, and primary production.

While the sedimentary record of dinoflagellates is intrinsically fragmentary and integrated in time, the reliability of past reconstructions is dependent on our knowledge of ecological processes occurring at short time-scales that capture seasonal and inter-

annual variability.

One of the most insightful approaches is the study of high-resolution sediment trap series coupled with hydrographic measurements. Such studies are rare, and often only possible in connection with long-term monitoring programmes and sustained collaborative efforts.

This manuscript reports on a detailed study of dinoflagellate cyst production over 2.5 years in the Cariaco Basin, strongly influenced by seasonal upwelling. The study has been well designed and executed. Comparing cyst production not only to environmental variables but also to biological indicators is novel and provides important insights into the trophic interactions of both individual taxa and groups. Mixotrophy is widespread in dinoflagellates, and the "classical" separation of dinoflagellates in two trophic groups that has been adopted by the paleo-community is clearly an oversimplification. This study clearly highlights this aspect, while also indicating different prey preferences within the heterotrophic group.

I congratulate the authors on providing detailed information for the cyst morphotypes encountered, as this will certainly be of use to future taxonomic work and sets a great example of good practice. Overall, this is a scientific contribution of excellent quality and importance, well within the scope of this journal, and I strongly recommend its publication.

Specific comments

Keywords: I suggest replacing the keyword "Harmful Algal Blooms" with "dinoflagellate cysts", to be consistent with the main focus of the study.

Page 2, Line 10 and Page 3, line 23 - The authors refer to several studies addressing anoxia in the basin, but it is not clear from the text whether anoxia is episodic or permanent (i.e. how frequent is deep water renewal?). Please clarify.

Page 4, Line 32; Page 5, Line 4 – Four sediment traps have been deployed at the

CARIACO station, but only Trap A (275m) was studied in terms of dinoflagellate cyst production. Why was this depth chosen? If sediments are also available from all the other traps, it would be extremely useful to study those as well, or at least to compare the shallowest with the deepest trap, in order to understand the dynamics of vertical cyst transport into the sediments. I hope the authors will consider doing this in the future.

Moreover, since this trap clogged during one of the most interesting events of the entire record, it would be important to investigate whether the other traps may provide a continuous record.

Page 5, line 23 – The palynological processing method used is rather standard, and since it is described in detailed, I found it confusing to refer to Pospelova et al. references, because at least in Pospelova et al. 2005, warm HF was used. If there is something specific in the method used by Pospelova et al. that ensures "optimal recovery" that should be mentioned in the text.

Results: I recommend the consistent use of past tense throughout the Results, especially considering that this trap is from the 1990's. E.g.: "The most abundant taxa were XXX", not are.

Page 12 – It is puzzling that as many as 24 extant taxa previously reported in the basin were not encountered in this study. This is mentioned, but not discussed. What are the possible explanations for this? Transport of cysts from elsewhere? Overwhelming dominance of Brigantedinium spp. masking the less dominant taxa (i.e. detection limit too low)? Or could it be that some species have cyst production cycles exceeding 2.5 years? I suggest discussing this intriguing observation further.

Page 13, Line 29 – This seems unlikely to me, because resting spores of Chaetoceros are, by definition, heavily silicified.

Page 15, Line 27 – I recommend referring to a more recent study, such as Jeong

et al. 2010. Jeong, H.J., Yoo, Y.D., Kim, J.S. et al. Ocean Sci. J. (2010) 45: 65. https://doi.org/10.1007/s12601-010-0007-2

Conclusions

Again in the conclusions, the use of present tense gives the reader the impression that assemblages as recorded in the 1990's in the trap are similar to present-day assemblages. Is there any information available that supports this? Can we be sure that the dinoflagellate cyst community of the Basin has not changed significantly over the past nearly 20 years (last trap sample dates from 1999)? I recommend carefully addressing this aspect throughout the text.

Technical corrections Terminology: Some terms are used in an inconsistent way. Consider your choice of: - Biogenic/Biogenous - Primary production/ Primary productivity - Planktic/ Planktonic - Biogenic silica: to my best knowledge, the correct abbreviation is BSi, not bioSi

Point-by-point suggestions:

Page 2, Line 13 – delete "at the site"; Line 16 – replace "the site of " by "under"; Line 21 – add "Here," before "We present"

Page 3, Line 2 – replace "accuracy" by "reliability".

I find it excessive to use 13 references here. It would be sufficient to refer to the first study ever, and then one study per main geographical area.

Page 4, Line 22 – The reader has already been introduced to station CARIACO, so this sentence can be simplified. Suggestion: ".... as part of the Cariaco Ocean Time-Series Program, at station CARIACO, located in the eastern Basin....", followed by "The programme has simultaneously produced oceanographic observations since 1995 (References)." The rest of the sentence is repeated elsewhere.

Page 5, Lines 2,3 – Consider changing to "... mounted on a carousel with a rotation

interval of 2 weeks".

Page 7, Line 22 – Simply writing ". . . cyst taxa and both physico-chemical and biological parameters" would flow better.

Page 9, Line 4 – "a six month-long" not "an"; Line 8 – "observed during the warmest intervals"; Line 17 - "that caused the trap to clog in April and May. . .." would flow better.; "Fluxes of biogenic material show" instead of share.

Page 10, Line 15 – "Over this time series, . . ." for simplification; Line 19 – "of" missing before "Echinidinium"; Line 30 – ". . ..towards the end".

Page 12, Line 20 – Not all the studies referred to are sediment trap studies. I suggest changing it to "consistently with studies from other upwelling systems"

Page 15, Line 1 – "from site to site" instead

Page 16 – ". . .when diatoms dominate primary. . .."

―――――――――――――――――

---

## Referee Comment (RC3) · Anonymous Referee #3 · 14 Mar 2018

The manuscript by Bringue and colleagues focuses on a short (2.5 years) time series of dinoflagellate cysts in the Cariaco Basin, a site well known for its remarkable sedimentary climatic record, and the home of the late CARIACO time-series program. The objectives of the work were to document the seasonal changes in dinoflagellate cyst production in the basin, to relate this production to climatic changes (e.g. upwelling, stratification, etc), and to investigate the relationships between dinoflagellates and other major planktonic groups that could impact cyst production/competition for resources. The authors state that the importance od this work is that it provides new insights into the ecology of cyst-producing dinoflagellates, and will allow for more detailed interpretations of fossil assemblages extracted from sedimentary records in the

basin and elsewhere.

The manuscript is interesting and in general well written; it does have however some things that need to be addressed. Details are provided below.

Pg. 2

Line 7: Replace "southern Caribbean upwelling system" for Southeastern Caribbean Sea, so it's not as redundant

Line 9: Please add Rueda-Roa, 2012 as citation for the secondary upwelling. Pg. 3

Line 3-5: Please pick the most relevant citations; there are too many.

Line 8: Please add 'subsequently' between 'has' and 'been' (e.g. . . .basin has subsequently been. . .) Pg 4

Citing figure 1 with location/map would have been helpful also in line 1 of pg 4, when reference is made to the CARIACO site.

Line 15-19: this has been stated in the introduction; I suggest removing/reducing the sentence in the introduction and leaving the longer description in the env. Setting section. Line 25: The CARIACO site is mentioned anew; I would suggest being more concise – the authors can choose whether to provide location in the setting section (top of pg. 4) or in the methods, but fragmented as it is now it's repetitive.

Pg. 8

Line 3-19: This does not seem to belong in the statistical analysis section; it should be moved to the beginning of the methods, to the 3.1 Sample collection and analyses part.

Results, section 4.1: Was there a reason why the authors decided to define their own upwelling/non-upwelling seasons, instead of following already defined 'seasons' from previously published literature? (e.g. Astor et al., 2013; Lorenzoni et al., 2011; Taylor

et al., 2012)? They mention their definition is consistent with others, then why not go with those?

Line 31: Please be quantitative; what does "higher temperatures in shallow waters" refer to? What is higher? What is shallow?

Pg 10

Line 19: Is there an 'of' missing between the words 'contribution Echinidinium'? Line 28: 'pulses' should be singular (pulse)

Pg. 12

Line 9-8: is repetitive, as has already been stated in the introduction

Line 23-27: Revise sentence and perhaps break it up; as it is, it's a bit confusing. Line 28: There is a 'are' missing at the end of the line, after 'sites'

Line 27-31: revise sentence; it's not clearly written and can be worded better Pg. 13

Line 14-15: Statistical significance is provided for correlations between fluxes and cyst, and the text suggests that the relationships are significant, though the provided suggests otherwise (p = 0.000). Generally, the p is set at 5% or 1%. The authors are advised to check their statistics and their interpretations.

The discussion in general presents many results which may be more appropriate to move to the 'results' section. For example, Pg. 16 has abundant results and references to figures which may be better moved to results, and the discussion section may then focus better on the actual discussion of results.

Pg. 17

Please revise correlation coefficients and p values – as they are it's impossible to tell whether they are significant or not.

The reference to the ENSO impact in the Strait of Georgia is out of context – the

geographic location is farther north and not even in the same ocean. It is suggested that it be removed as it adds nothing to the discussion.

Pg 19

Line 1: The authors conclude that "On interannual time scales, dinoflagellate cyst production seems to be influenced by the strong 1997/98 El Niño event, with a one year lag", though from their data and discussion it was apparent that they were not able to draw this conclusion?

It is also unclear how the "work expands our knowledge of cyst-producing dinoflagellate ecology, helping the interpretation of fossil assemblages from the basin's sedimentary record and worldwide." It would have been a benefit if the authors had included in the discussion a paragraph where they tied it all together and specifically explained how this work would help the interpretation of fossil assemblages from the basin's sedimentary record. The authors stress the importance of the work in the introduction, but then limit themselves at characterizing the cysts and seasonality and don't put the results in the context of why this is important.

---

## Author Response (AR1)

Dinoflagellate cysts are increasingly used for marine paleoenvironmental interpretations, based on "cyst signals" recognized from studies of cyst distributions in the present-day ocean. These include signals for dissolved nutrients, SST, salinity, and distance from shore, largely developed by comparing cyst assemblages in surface sediments with environmental conditions in correspondingly overlying surface waters. The accuracy of signals inferred from such comparisons is dependent on the level of our understanding of which cysts are formed where and when in the ecological system. There are obvious limitations to simply comparing cysts in bottom sediments

with overlying surface water: cysts may be transported significant distances from their point of origin in the plankton prior to being deposited in the sediment (especially in deep oceanic or otherwise turbulent waters), possibly obscuring evidence of where they were formed; the age of cysts in bottom sediments is dependent on rates of sedimentation, potentially obscuring when the cysts were formed. The hydrographic data available for comparison with the sediments are of varied quality. Sediment traps as deployed here can provide more plausible information, while also recognizing certain limitations to this approach, too, as the authors do. By choosing the upper trap of this array (at 275m), should reduce the possibility for long-transport of the cysts sampled, and the succession of cup- collections is expected to show the approximate time of cyst formations in the upper water mass. The series of hydrographic data collected at the same station and times should allow close cyst/environment comparisons. The results produced seem to confirm a by now well established nutrient signal with heterotrophic species dominating the cyst signal with increased nutrients from upwelling – with some interesting first suggestions of nuances involving different species with differing degrees of upwelling. This is excellent work that pushes our understanding a step further by suggesting when and where the reported cysts may have formed, allowing this to be correlated to the corresponding environmental parameters, in at least this one location. Furthermore, including data for other plankton groups has allowed some suggestions of possible links to the very complex marine food web. This work sets the bar high for the many more such studies that will be needed to increase our understanding of just how precise the cyst signals may be. As a reviewer, I could wish that more studies were as well designed, carried out and reported as this, and the authors are to be congratulated. The only possible improvements I could suggest for the paper would be to indicate any currents affecting the water entering the basin (possible cyst transport?), and to indicate if possible what proportion of the cysts trapped had "fresh" cell contents. Our experience with sediment traps is that cell contents may be good indicators of "newly-formed" cysts. The authors perhaps wisely avoid serious attempts to relate this evidence from one trap depth to the bottom sediments that form

the broader reason for carrying out such studies, but I hope they will continue to investigate the lower trap samples eventually to see how accurately the cyst signal they identify is translated into the sediment.

[Figure]

**Author Comments on "Physico-chemical and biological factors influencing dinoflagellate cyst production in the Cariaco Basin", by Manuel Bringué et al. (bg-2017-497)**

**Response to Referee 1 (B. Dale)**

The authors wish to deeply thank B. Dale for his very positive and constructive review of the manuscript. The referee raised three points that will be addressed below:

1- Suggestion '*to indicate any currents affecting the water entering the basin (possible cyst transport?)*'

We (the authors) agree that the manuscript would benefit from adding a statement about currents entering the basin that can potentially transport cysts. The following statement will be included in Section 2 of the revised manuscript: "Observations (and models) of horizontal velocities of currents in the basin indicate that horizontal transport is weak in the upper water column (U and V components < 0.2 m s$^{-1}$) and negligible below sill depth (Alvera-Azcárate et al. [2009], *Ocean Dynamics* 59:97–120; see their Fig. 6). The potential for horizontal transport is thus believed to be minimal at the CARIACO station." In addition, as the referee mentioned, the use of Trap A samples (shallow trap at 275 m depth) also reduces the possibility for long cyst transport.

2- Suggestion '*to indicate if possible what proportion of the cysts trapped had "fresh" cell contents.*'

While we report the proportions of cysts with cell content in Fig. 5, it is true that the overall proportion of such cysts *vs* empty ones is not stated. The main reason is that this constitutes an aspect that is worth investigating in detail and will be done in a follow up study. Given the focus of the present study, we found limited relevance in discussing the proportion of cysts with cell content, because 1- the potential for cyst transport is minimal in this setting, and 2- the signal did not seem to show any particular seasonality for any given cyst taxon (i.e., the proportion of cysts with cell content varied by species, but for each taxon the signal was rather consistent throughout the time series). This will be added to Section 4.3 in the revised manuscript.

3- The referee hopes the authors '*will continue to investigate the lower trap samples eventually to see how accurately the cyst signal they identify is translated into the sediment*'

Our next study will focus on this specific issue, in the Cariaco Basin as well as in other coastal settings.

Biogeosciences Discuss.,
https://doi.org/10.5194/bg-2017-497-RC2, 2018

[Figure]

Dinoflagellates cysts are ubiquitous in coastal ecosystems worldwide, and have been increasingly used in palaeoenvironmental studies as indicators of past changes in e.g. sea surface temperature, salinity, nutrient status, and primary production.

While the sedimentary record of dinoflagellates is intrinsically fragmentary and integrated in time, the reliability of past reconstructions is dependent on our knowledge of ecological processes occurring at short time-scales that capture seasonal and inter-

annual variability.

One of the most insightful approaches is the study of high-resolution sediment trap series coupled with hydrographic measurements. Such studies are rare, and often only possible in connection with long-term monitoring programmes and sustained collaborative efforts.

This manuscript reports on a detailed study of dinoflagellate cyst production over 2.5 years in the Cariaco Basin, strongly influenced by seasonal upwelling. The study has been well designed and executed. Comparing cyst production not only to environmental variables but also to biological indicators is novel and provides important insights into the trophic interactions of both individual taxa and groups. Mixotrophy is widespread in dinoflagellates, and the "classical" separation of dinoflagellates in two trophic groups that has been adopted by the paleo-community is clearly an oversimplification. This study clearly highlights this aspect, while also indicating different prey preferences within the heterotrophic group.

I congratulate the authors on providing detailed information for the cyst morphotypes encountered, as this will certainly be of use to future taxonomic work and sets a great example of good practice. Overall, this is a scientific contribution of excellent quality and importance, well within the scope of this journal, and I strongly recommend its publication.

Specific comments

Keywords: I suggest replacing the keyword "Harmful Algal Blooms" with "dinoflagellate cysts", to be consistent with the main focus of the study.

Page 2, Line 10 and Page 3, line 23 - The authors refer to several studies addressing anoxia in the basin, but it is not clear from the text whether anoxia is episodic or permanent (i.e. how frequent is deep water renewal?). Please clarify.

Page 4, Line 32; Page 5, Line 4 – Four sediment traps have been deployed at the

[Figure]

CARIACO station, but only Trap A (275m) was studied in terms of dinoflagellate cyst production. Why was this depth chosen? If sediments are also available from all the other traps, it would be extremely useful to study those as well, or at least to compare the shallowest with the deepest trap, in order to understand the dynamics of vertical cyst transport into the sediments. I hope the authors will consider doing this in the future.

Moreover, since this trap clogged during one of the most interesting events of the entire record, it would be important to investigate whether the other traps may provide a continuous record.

Page 5, line 23 – The palynological processing method used is rather standard, and since it is described in detailed, I found it confusing to refer to Pospelova et al. references, because at least in Pospelova et al. 2005, warm HF was used. If there is something specific in the method used by Pospelova et al. that ensures "optimal recovery" that should be mentioned in the text.

Results: I recommend the consistent use of past tense throughout the Results, especially considering that this trap is from the 1990's. E.g.: "The most abundant taxa were XXX", not are.

Page 12 – It is puzzling that as many as 24 extant taxa previously reported in the basin were not encountered in this study. This is mentioned, but not discussed. What are the possible explanations for this? Transport of cysts from elsewhere? Overwhelming dominance of Brigantedinium spp. masking the less dominant taxa (i.e. detection limit too low)? Or could it be that some species have cyst production cycles exceeding 2.5 years? I suggest discussing this intriguing observation further.

Page 13, Line 29 – This seems unlikely to me, because resting spores of Chaetoceros are, by definition, heavily silicified.

Page 15, Line 27 – I recommend referring to a more recent study, such as Jeong

et al. 2010. Jeong, H.J., Yoo, Y.D., Kim, J.S. et al. Ocean Sci. J. (2010) 45: 65. https://doi.org/10.1007/s12601-010-0007-2

Conclusions

Again in the conclusions, the use of present tense gives the reader the impression that assemblages as recorded in the 1990's in the trap are similar to present-day assemblages. Is there any information available that supports this? Can we be sure that the dinoflagellate cyst community of the Basin has not changed significantly over the past nearly 20 years (last trap sample dates from 1999)? I recommend carefully addressing this aspect throughout the text.

Technical corrections Terminology: Some terms are used in an inconsistent way. Consider your choice of: - Biogenic/Biogenous - Primary production/ Primary productivity - Planktic/ Planktonic - Biogenic silica: to my best knowledge, the correct abbreviation is BSi, not bioSi

Point-by-point suggestions:

Page 2, Line 13 – delete "at the site"; Line 16 – replace "the site of " by "under"; Line 21 – add "Here," before "We present"

Page 3, Line 2 – replace "accuracy" by "reliability".

I find it excessive to use 13 references here. It would be sufficient to refer to the first study ever, and then one study per main geographical area.

Page 4, Line 22 – The reader has already been introduced to station CARIACO, so this sentence can be simplified. Suggestion: ".... as part of the Cariaco Ocean Time-Series Program, at station CARIACO, located in the eastern Basin....", followed by "The programme has simultaneously produced oceanographic observations since 1995 (References)." The rest of the sentence is repeated elsewhere.

Page 5, Lines 2,3 – Consider changing to "... mounted on a carousel with a rotation

interval of 2 weeks".

Page 7, Line 22 – Simply writing ". . . cyst taxa and both physico-chemical and biological parameters" would flow better.

Page 9, Line 4 – "a six month-long" not "an"; Line 8 – "observed during the warmest intervals"; Line 17 - "that caused the trap to clog in April and May. . .." would flow better.; "Fluxes of biogenic material show" instead of share.

Page 10, Line 15 – "Over this time series, . . ." for simplification; Line 19 – "of" missing before "Echinidinium"; Line 30 – ". . ..towards the end".

Page 12, Line 20 – Not all the studies referred to are sediment trap studies. I suggest changing it to "consistently with studies from other upwelling systems"

Page 15, Line 1 – "from site to site" instead

Page 16 – ". . .when diatoms dominate primary. . .."

————————————————

[Figure]

**Author Comments on "Physico-chemical and biological factors influencing dinoflagellate cyst production in the Cariaco Basin", by Manuel Bringué et al. (bg-2017-497)**

**Response to Referee 2 (Dr. Ribeiro)**

The authors are very grateful for Dr. Ribeiro's positive and thoughtful comments, which validate the research featured in this manuscript while helping improve its content. The authors' response to the referee's specific comments, are detailed below in the order they were listed:

- *Keyword 'Harmful Algal Blooms'*: The purpose of this keyword is to ensure that researchers interested in Harmful Algal Blooms are made aware of this study because of the record it provides on potentially harmful taxa, even though it is indeed a relatively minor component in the study. Since the suggested keyword 'dinoflagellate cysts' is already stated in the title, we think it is best to retain 'Harmful Algal Blooms' in the list of Keywords.

- *'Episodic or permanent anoxia'*: We agree that this needs to be specified and 'permanent' will be added at both places in the text (Page 2, Line 10 and Page 3, Line 23). To be accurate, ventilation events that are discussed in the papers cited refer to intrusion of waters down to ~ 320 m depth (e.g., Astor et al. 2003, Continental Shelf Research 23, 125-144), leaving the rest of the water column (i.e., down to 1400 m) under fully anoxic conditions. Note that 'permanent' anoxic conditions refer only to the Holocene (as well as other interglacials), since there is evidence in the sedimentary record of the basin that bottom waters were oxygenated during the last glacial (e.g., Peterson et al., 1991, Paleoceanography 6, 99-119); however, given the time scale of the present study and for clarity, we will simply specify 'permanent' in the text.

- *Considering analyzing the sediment trap content in all traps from the deployment*: We are considering doing this work in the nearest future.

- *Using the record from other (deeper) traps to fill gaps in the Trap A record*: This is something the authors considered, as records from Trap A and B have been showed to yield comparable mass fluxes (e.g., Thunell et al. 2000, Limnology and Oceanography 45, 300-308) and were used interchangeably in the study of other microfossil groups at the site (e.g., Romero et al., 2009, Deep-Sea Research Part I-Oceanographic Research Papers 56, 571-581). Samples from deeper traps, however, typically yield very little material (Thunell et al., 2000, op. cit.) and may be affected by turbidity flows (Thunell et al., 1999, Nature 398, 233-236) and were thus were not deemed suitable for analyses. Unfortunately, for the duration of the time series of the present study, no sample from Trap B were available to complement the Trap A record. This will be clarified in the text.

- *Palynological processing method*: We respectfully disagree with the referee on the idea that the palynological processing method is 'rather standard'. While there are large similarities between methods used in different labs (e.g., repeated wet sievings, use of HCl and HF), there are particular steps in the method that are not necessarily standardized, especially when it comes to processing sediment trap samples (for instance, timing of sonication, limited time the samples are exposed to HCl, etc.). It is worth noting that the wording of 'warm' HF in Pospelova et al. (2005) was meant to be read as 'room temperature' (as opposed to 'refrigerator-cold'), and we will thus change the reference to Pospelova et al. (2010) to avoid any confusion.

- *Use of past tense in Results*: We agree that the use of past tense is more appropriate to relate changes that occurred at the time of sample collection. This will be addressed in the revised version of the manuscript.

- *About the 24 extant taxa previously reported in the basin that were not encountered in this study*: we believe that the most likely explanation for the absence of these 24 cyst taxa in our 2.5 year-long record is because these taxa were documented from sediments spanning the last ~ 73,000 years, as stated in the text (parenthesis before the enumeration of taxa). Most of these taxa are quite rare in the Cariaco Basin sedimentary record (see Gonzalez et al. 2008, Paleoceanography 23, and Mertens et al., 2009, Boreas 38, 647-662) and it is not surprising that they were not captured during the short window (2.5 years) of our survey.

- *Chaetoceros resting spores*: This comment refers to the statement: '(…) the large increases in diatom fluxes recorded in 1999 are mainly attributable to small and/or weakly silicified species, namely *Cyclotella litoralis* and resting spores of *Chaetoceros* (Romero et al., 2009), which may not result in elevated mass fluxes of bioSi.' We recognize that the wording is confusing, and the statement will be changed in the text to: '(…) the large increases in diatom fluxes recorded in 1999 are mainly attributable to small (e.g., resting spores of *Chaetoceros*) and/or weakly silicified (*Cyclotella litoralis*) species (Romero et al., 2009), which may not result in elevated mass fluxes of bioSi.'

- *Reference to Jeong et al. 2010*: We thank the referee for bringing this to our attention. The reference to this review will be added here (Page 15 Line 27).

- *Use of past tense in the Conclusion*: We agree that using the past tense is more appropriate here as well; this should and will be corrected in the revised version of the manuscript.

*- Technical corrections (terminology)*:

1. *Biogenic/biogenous*. The word 'biogenic' will be used everywhere for consistency in the revised version of the manuscript.

2. *Primary production/productivity*. We thank the referee for bringing this point to our attention. However, what may appear as an inconsistency actually refers to different aspects, with 'production' generally used with quantities in mind (either masses or rates), and 'productivity' referring more to the process itself.

3. *Planktic/planktonic*. 'Planktonic' is already used everywhere in the text.

4. *Biogenic silica*: The use of 'bioSi' as an abbreviation for 'biogenic silica' (as opposed to 'BSi') is a matter of preference and both are routinely found in the literature. To the best of our knowledge, there is no formally accepted convention.

*- Point-by-point suggestions (Pages 2 and 3)*: All suggested changes will be implemented.

*- 13 citations on Page 3, Lines 3-5*: We respectfully disagree with the referee on this point, as we find it important to acknowledge previous work that set the ground for this study and there are few redundancies in terms of geographical areas.

*- Point-by-point suggestions (Page 4)*: We thank the referee for bringing this to our attention; however, we prefer to keep the wording as it is, because it allows the first 2 sentences of the Material and Methods section to rapidly summarise the sampling program (despite the bit of repetition elsewhere), an important aspect for a reader who is interested into this aspect of the study.

*- Point-by-point suggestions (Pages 5, 7)*: The referee's suggestions should and will be implemented in the revised text.

*- Point-by-point suggestion (Pages 9, Line 4)*: 'An' will be retained since the symbol '~' stands for 'approximately'.

*- Point-by-point suggestions (Page 9, Lines 8, 17)*: The suggested wording is better and will be implemented. However, the use of 'share' will be retained (instead of 'show') as it indicates the common pattern of variation between biogenous material and total mass fluxes.

*- Point-by-point suggestions (Page 10, Line 15)*: We think that the original wording of 'Over the duration of the time series' should be kept for accuracy.

- *Point-by-point suggestions (Page 10, Lines 19, 30)*: The suggestions will be implemented, thank you for noticing.

- *Point-by-point suggestions (Pages 12, 15, 16)*: The suggestions will be implemented, many thanks!

Biogeosciences Discuss.,
https://doi.org/10.5194/bg-2017-497-RC3, 2018

[Figure]
The manuscript by Bringue and colleagues focuses on a short (2.5 years) time series of dinoflagellate cysts in the Cariaco Basin, a site well known for its remarkable sedimentary climatic record, and the home of the late CARIACO time-series program. The objectives of the work were to document the seasonal changes in dinoflagellate cyst production in the basin, to relate this production to climatic changes (e.g. upwelling, stratification, etc), and to investigate the relationships between dinoflagellates and other major planktonic groups that could impact cyst production/competition for resources. The authors state that the importance od this work is that it provides new insights into the ecology of cyst-producing dinoflagellates, and will allow for more detailed interpretations of fossil assemblages extracted from sedimentary records in the

basin and elsewhere.

The manuscript is interesting and in general well written; it does have however some things that need to be addressed. Details are provided below.

Pg. 2

Line 7: Replace "southern Caribbean upwelling system" for Southeastern Caribbean Sea, so it's not as redundant

Line 9: Please add Rueda-Roa, 2012 as citation for the secondary upwelling. Pg. 3

Line 3-5: Please pick the most relevant citations; there are too many.

Line 8: Please add 'subsequently' between 'has' and 'been' (e.g. . . .basin has subsequently been. . .) Pg 4

Citing figure 1 with location/map would have been helpful also in line 1 of pg 4, when reference is made to the CARIACO site.

Line 15-19: this has been stated in the introduction; I suggest removing/reducing the sentence in the introduction and leaving the longer description in the env. Setting section. Line 25: The CARIACO site is mentioned anew; I would suggest being more concise – the authors can choose whether to provide location in the setting section (top of pg. 4) or in the methods, but fragmented as it is now it's repetitive.

Pg. 8

Line 3-19: This does not seem to belong in the statistical analysis section; it should be moved to the beginning of the methods, to the 3.1 Sample collection and analyses part.

Results, section 4.1: Was there a reason why the authors decided to define their own upwelling/non-upwelling seasons, instead of following already defined 'seasons' from previously published literature? (e.g. Astor et al., 2013; Lorenzoni et al., 2011; Taylor

et al., 2012)? They mention their definition is consistent with others, then why not go with those?

Line 31: Please be quantitative; what does "higher temperatures in shallow waters" refer to? What is higher? What is shallow?

Pg 10

Line 19: Is there an 'of' missing between the words 'contribution Echinidinium'? Line 28: 'pulses' should be singular (pulse)

Pg. 12

Line 9-8: is repetitive, as has already been stated in the introduction

Line 23-27: Revise sentence and perhaps break it up; as it is, it's a bit confusing. Line 28: There is a 'are' missing at the end of the line, after 'sites'

Line 27-31: revise sentence; it's not clearly written and can be worded better Pg. 13

Line 14-15: Statistical significance is provided for correlations between fluxes and cyst, and the text suggests that the relationships are significant, though the provided suggests otherwise (p = 0.000). Generally, the p is set at 5% or 1%. The authors are advised to check their statistics and their interpretations.

The discussion in general presents many results which may be more appropriate to move to the 'results' section. For example, Pg. 16 has abundant results and references to figures which may be better moved to results, and the discussion section may then focus better on the actual discussion of results.

Pg. 17

Please revise correlation coefficients and p values – as they are it's impossible to tell whether they are significant or not.

The reference to the ENSO impact in the Strait of Georgia is out of context – the

geographic location is farther north and not even in the same ocean. It is suggested that it be removed as it adds nothing to the discussion.

Pg 19

Line 1: The authors conclude that "On interannual time scales, dinoflagellate cyst production seems to be influenced by the strong 1997/98 El Niño event, with a one year lag", though from their data and discussion it was apparent that they were not able to draw this conclusion?

It is also unclear how the "work expands our knowledge of cyst-producing dinoflagellate ecology, helping the interpretation of fossil assemblages from the basin's sedimentary record and worldwide." It would have been a benefit if the authors had included in the discussion a paragraph where they tied it all together and specifically explained how this work would help the interpretation of fossil assemblages from the basin's sedimentary record. The authors stress the importance of the work in the introduction, but then limit themselves at characterizing the cysts and seasonality and don't put the results in the context of why this is important.

[Figure]

**Author Comments on "Physico-chemical and biological factors influencing dinoflagellate cyst production in the Cariaco Basin", by Manuel Bringué et al. (bg-2017-497)**

**Response to Referee 3 (Anonymous)**

The authors wish to thank Referee 3 for her/his thoughtful critical review which will help improve the manuscript. Our response to each point raised by the referee is detailed below:

- *Pg. 2 Line 7: Replace "southern Caribbean upwelling system" for Southeastern Caribbean Sea, so it's not as redundant*: This will be implemented in the revised version of the manuscript.

- *Pg. 2 Line 9: Please add Rueda-Roa, 2012 as citation for the secondary upwelling*: This will be implemented, even though this references to an unpublished or peer-reviewed - PhD thesis. Thank you for providing this reference.

- *Pg. 3 Line 3-5: Please pick the most relevant citations; there are too many*: We respectfully disagree with the referee on this, as we find it important to acknowledge previous work that set the ground for this study.

- *Pg. 3 Line 8: Please add 'subsequently' between 'has' and 'been'*: This will be implemented.

- *Pg. 4 Citing figure 1 with location/map would have been helpful also in line 1 of pg 4, when reference is made to the CARIACO site*: The reference to Fig. 1 will be added.

- *Pg. 4 Line 15-19: this has been stated in the introduction; I suggest removing/reducing the sentence in the introduction and leaving the longer description in the Env. Setting section*: Thank you for bringing this to our attention; however, we think it is appropriate to mention the presence of laminated sediments in the basin both in the Introduction (where it highlights the relevance of this calibration effort given the basin's potential for paleoreconstructions) and the Environmental Setting section (where the question of how surface processes affect the nature of the sediments should be detailed).

- *Pg. 4 Line 25: The CARIACO site is mentioned anew; I would suggest being more concise – the authors can choose whether to provide location in the setting section (top of pg. 4) or in the methods, but fragmented as it is now it's repetitive*: We respectfully disagree with the referee on this point, as we think station CARIACO needs to be mentioned in both sections. The mention in the Environmental Setting section is necessary because we need to be specific about the location of measured parameters (e.g., primary productivity) mentioned in the text, and it is also necessary in the Material and Methods section for the specific location (with coordinates) of the sediment trap mooring and water column sampling site.

*- Pg. 8 Line 3-19: This does not seem to belong in the statistical analysis section; it should be moved to the beginning of the methods, to the 3.1 Sample collection and analyses part*: The physico-chemical and biological parameters mentioned in Lines 3-19 were mostly extracted from previously published literature (thus not new in this study) and were used exclusively in multivariate statistical analyses. Thus, we find it more appropriate to retain the text in Section 3.4.

*- Results, section 4.1: Was there a reason why the authors decided to define their own upwelling/non-upwelling seasons, instead of following already defined 'seasons' from previously published literature? (e.g. Astor et al., 2013; Lorenzoni et al., 2011; Taylor et al., 2012)? They mention their definition is consistent with others, then why not go with those?* This is an excellent point which was discussed extensively within co-authors and with other colleagues who are involved in the Cariaco Basin Ocean Time Series and/or familiar with the hydrography of the basin. The 'already defined seasons' in the papers mentioned by the referee relate to general trends (e.g., upwelling season from January – May, non-upwelling from June – November, plus secondary upwelling events) and failed to capture the detailed dynamics of the system at the time scale of interest (and resolution) for this study. All the papers mentioned above also acknowledge that upwelling varies greatly between years. The upwelling index provided in Astor et al. (2013; their Fig. 2C) provided a much more detailed picture of upwelling dynamics but was derived entirely from wind data. Consensus was reached upon adopting this definition of upwelling because it relies on in-situ water column measurements and it captures the timing of upwelling events in much more detail, which was paramount for the interpretation of dinoflagellate cyst data. It was also agreed that providing a lengthy discussion on this particular point in the manuscript was unnecessary, as the definitions of respective upwelling intervals are clearly defined.

*- Line 31: Please be quantitative; what does "higher temperatures in shallow waters" refer to? What is higher? What is shallow?* This sentence is meant to give a general overview of variations in temperatures in the upper 100 m of the water column. The specific values of SST are stated ('between ~ 19 and 28 °C'). The words 'in shallow waters' will be replaced by 'close to the surface' to avoid confusion.

*- Pg 10 Line 19: Is there an 'of' missing between the words 'contribution Echinidinium'?* This will be addressed, thanks for noticing.

*- Pg 10 Line 28: 'pulses' should be singular (pulse)*: Indeed, this will be changed.

*- Pg. 12 Line 9-8: is repetitive, as has already been stated in the introduction*: It is actually necessary to mention these studies here again as the paragraph is about comparing the cyst record of the trap time series (this study) with previously reported taxa from the basin's sedimentary record.

*- Pg. 12 Line 23-27: Revise sentence and perhaps break it up; as it is, it's a bit confusing*: Thanks for pointing this out, it will be addressed in the manuscript by rewording and splitting the sentence.

*- Pg. 12 Line 28: There is a 'are' missing at the end of the line, after 'sites'*: This section of the sentence between commas is part of an enumeration of characteristics shared by the Cariaco and Santa Barbara basins that does not need a verb - the actual conjugated verb in this sentence is found later at Line 30 ('…cyst fluxes *are* about five times lower…').

*- Pg. 12 Line 27-31: revise sentence; it's not clearly written and can be worded better*: We acknowledge that the sentence is rather long but we believe it is the most effective way to provide a comparison between sites (Santa Barbara and Cariaco basins) and cyst records, without making the text heavy.

*- Pg. 13 Line 14-15: Statistical significance is provided for correlations between fluxes and cyst, and the text suggests that the relationships are significant, though the provided suggests otherwise (p = 0.000). Generally, the p is set at 5% or 1%. The authors are advised to check their statistics and their interpretations*: The notation 'p = 0.000' actually means that p is well below the typical thresholds of either 5% (= 0.05) or 1 % (= 0.01). In statistics, 'p' refers to the probability determined by the test, which is then compared to a threshold value (usually 5% or 1%); if p is inferior to the threshold value, one can safely reject the null hypothesis ($H_0$), and safely accept the alternative hypothesis ($H_1$). We thank the referee for ensuring the statistics are sound, but we find nothing to change. We will change 'p = 0.000' to 'p < 0.001' to avoid confusion.

*- The discussion in general presents many results which may be more appropriate to move to the 'results' section. For example, Pg. 16 has abundant results and references to figures which may be better moved to results, and the discussion section may then focus better on the actual discussion of results*: We thank the referee for her/his suggestion, but we find it appropriate to retain the current organisation. While some statements in Section 5.2 appear to present information not explicitly detailed in the Results, they are necessary to discuss the interactions between dinoflagellates and other major planktonic groups and are only relevant in this section. We have considered moving some of the content to Results, but found it more appropriate in the Discussion. Note that this difference in style may arise from different practices observed in varying fields of study.

*- Pg. 17 Please revise correlation coefficients and p values – as they are it's impossible to tell whether they are significant or not*. All p-values mentioned being smaller than 0.05, all correlation coefficients stated in the text are statistically significant. We have considered changing the text to something like 'SST × NAO: Pearson's *r* = -0.33, p = 0.011; SST × MEI: r = -0.28, p = 0.031' but found it to be more

confusing. The sentence of Pg. 17 Lines 15-17 will be reworded to specify that each correlation is significant.

*- The reference to the ENSO impact in the Strait of Georgia is out of context – the geographic location is farther north and not even in the same ocean. It is suggested that it be removed as it adds nothing to the discussion.* We respectfully disagree with the referee on this point, as the Strait of Georgia record is the ONLY other record of a possible influence of ENSO on dinoflagellate cyst production in the world. This absolutely needs to be discussed here, despite the relatively distant geographical locations.

*- Pg 19 Line 1: The authors conclude that "On interannual time scales, dinoflagellate cyst production seems to be influenced by the strong 1997/98 El Niño event, with a one year lag", though from their data and discussion it was apparent that they were not able to draw this conclusion?* The study makes a case that clear impacts of ENSO on dinoflagellate cyst production are not detected when no time-lag is applied. However, considering a one year lag (as suggested in Enfield and Mayer, 1997; Giannini et al., 2001), the data strongly suggest that cyst production is negatively impacted by the strong 1997/98 El Niño event, a response that is compared with other records of such impacts on the lower trophic levels from various locations in Section 5.3. The only limitation is that due to the relatively short duration of the sediment trap time series (2.5 years), there are not enough data points to compute robust and meaningful correlations with the ENSO index. The statement (Pg. 19 Line 1) is carefully worded and it is also clearly stated that longer time series are needed to properly investigate ENSO effects' (Pg. 18 Line 7).

*- It is also unclear how the "work expands our knowledge of cyst-producing dinoflagellate ecology, helping the interpretation of fossil assemblages from the basin's sedimentary record and worldwide." It would have been a benefit if the authors had included in the discussion a paragraph where they tied it all together and specifically explained how this work would help the interpretation of fossil assemblages from the basin's sedimentary record. The authors stress the importance of the work in the introduction, but then limit themselves at characterizing the cysts and seasonality and don't put the results in the context of why this is important.* We appreciate the referee's comment which reflects her/his care to ensure the authors keep a larger perspective in mind. While we acknowledge that someone outside the field might expect more on the interpretation of fossil assemblages, such a discussion would be out of the scope of this study. The introduction clearly makes a case that such sediment trap studies constitute a 'calibration' effort, which in turn help in the interpretation of fossil assemblages. It is the intention of the authors to continue investigating the dinoflagellate cyst sedimentary record in the basin (and elsewhere), and the findings of the present study will be extremely valuable at that time.

[revised manuscript text omitted]

Sample information and flux data (the legible portion):

| USC ID | U/c ID | Cup opens | Duration (days) | Total dino cyst flux | Total autotroph flux | Total heterotroph flux | Total dino cysts counted |
|---|---|---|---|---|---|---|---|
| 3A-1 | 15-789 | 08-Nov-96 | 14 | 28035 | 557 | 27478 | 304 |
| 3A-2 | 15-790 | 22-Nov-96 | 14 | 28241 | 445 | 27617 | 317 |
| 3A-3 | 15-791 | 06-Dec-96 | 14 | 4306 | 55 | 4250 | 312 |
| 3A-4 | 15-792 | 20-Dec-96 | 14 | 9706 | 204 | 9473 | 216 |
| 3A-5 | 15-793 | 03-Jan-97 | 14 | 16609 | 259 | 16350 | 321 |
| 3A-6 | 15-794 | 17-Jan-97 | 14 | 18488 | 229 | 18259 | 333 |
| 3A-7 | 15-795 | 31-Jan-97 | 14 | 17352 | 202 | 17150 | 323 |
| 3A-8 | 15-796 | 14-Feb-97 | 14 | 55676 | 323 | 55323 | 344 |
| 3A-9 | 15-797 | 28-Feb-97 | 14 | 49520 | 152 | 49368 | 315 |
| 3A-10 | 15-798 | 14-Mar-97 | 14 | 69188 | 969 | 68219 | 325 |
| 4A-1 | 15-799 | 15-May-97 | 7 | 58007 | 4001 | 54007 | 357 |
| 4A-2 | 15-800 | 22-May-97 | 14 | 9900 | 1917 | 7983 | 319 |
| 4A-3 | 15-801 | 05-Jun-97 | 14 | 6873 | 286 | 6564 | 315 |
| 4A-4 | 15-802 | 19-Jun-97 | 14 | 10733 | 690 | 10011 | 312 |
| 4A-5 | 15-803 | 03-Jul-97 | 14 | 43048 | 7175 | 35874 | 342 |
| 4A-6 | 15-804 | 17-Jul-97 | 14 | 40816 | 12688 | 28127 | 378 |
| 4A-7 | 15-805 | 31-Jul-97 | 14 | 34311 | 4480 | 29832 | 337 |
| 4A-8 | 15-806 | 14-Aug-97 | 14 | 8489 | 1714 | 6722 | 317 |
| 4A-9 | 15-807 | 28-Aug-97 | 14 | 13394 | 1680 | 11714 | 311 |
| 4A-10 | 15-808 | 11-Sep-97 | 14 | 14288 | 1324 | 12873 | 313 |
| 4A-11 | 15-809 | 25-Sep-97 | 14 | 6738 | 1685 | 5054 | 312 |
| 4A-12 | 15-810 | 09-Oct-97 | 14 | 3117 | 359 | 2729 | 321 |
| 4A-13 | 15-811 | 23-Oct-97 | 11 | 3703 | 498 | 3113 | 320 |
| 5A-1 | 15-812 | 13-Nov-97 | 7 | 10172 | 1014 | 8831 | 311 |
| 5A-2 | 15-813 | 20-Nov-97 | 14 | 8581 | 545 | 7955 | 315 |
| 5A-3 | 15-814 | 04-Dec-97 | 14 | 21182 | 665 | 20497 | 318 |
| 5A-4 | 15-815 | 18-Dec-97 | 14 | 11082 | 781 | 10301 | 312 |
| 5A-5 | 15-816 | 01-Jan-98 | 14 | 30723 | 569 | 30059 | 324 |
| 5A-6 | 15-817 | 15-Jan-98 | 14 | 1569 | 28 | 1541 | 281 |
| 5A-7 | 15-818 | 29-Jan-98 | 14 | 4543 | 303 | 4211 | 315 |
| 5A-8 | 15-819 | 12-Feb-98 | 14 | 4167 | 268 | 3899 | 311 |
| 5A-9 | 15-820 | 26-Feb-98 | 14 | 14760 | 596 | 14164 | 322 |
| 5A-10 | 15-821 | 12-Mar-98 | 14 | 19509 | 246 | 19263 | 317 |
| 5A-11 | 15-822 | 26-Mar-98 | 14 | 39694 | 1623 | 38071 | 318 |
| 5A-12 | 15-823 | 09-Apr-98 | 14 | 13896 | 438 | 13458 | 317 |
| 5A-13 | 15-824 | 23-Apr-98 | 12 | 11884 | 606 | 11278 | 314 |
| 6A-1+2 | 15-825+6 | 03-Jun-98 | 7+7 | 4819 | 744 | 4075 | 314 |
| 6A-3+4 | 15-827+8 | 17-Jun-98 | 7+7 | 7248 | 361 | 6887 | 313 |
| 6A-5 | 15-829 | 01-Jul-98 | 14 | 23057 | 757 | 22300 | 335 |
| 6A-6 | 15-830 | 15-Jul-98 | 14 | 8739 | 246 | 8439 | 320 |
| 6A-7 | 15-831 | 29-Jul-98 | 14 | 9689 | 679 | 8979 | 314 |
| 6A-8 | 15-832 | 12-Aug-98 | 14 | 7788 | 195 | 7592 | 319 |
| 6A-9 | 15-833 | 26-Aug-98 | 14 | 5862 | 203 | 5591 | 260 |
| 6A-10 | 15-834 | 09-Sep-98 | 14 | 5148 | 223 | 4925 | 254 |
| 6A-11 | 15-835 | 23-Sep-98 | 14 | 9733 | 188 | 9451 | 311 |
| 6A-12 | 15-836 | 07-Oct-98 | 14 | 6102 | 196 | 5867 | 311 |
| 6A-13 | 15-837 | 21-Oct-98 | 12 | 4275 | 90 | 4149 | 238 |
| 7A-1 | 15-838 | 07-Nov-98 | 14 | 2106 | 100 | 2005 | 188 |
| 7A-2 | 15-839 | 21-Nov-98 | 14 | 19867 | 123 | 19440 | 126 |
| 7A-3 | 15-840 | 05-Dec-98 | 14 | 13264 | 287 | 12938 | 319 |
| 7A-4 | 15-841 | 19-Dec-98 | 14 | 1680 | 36 | 1637 | 323 |
| 7A-5 | 15-842 | 02-Jan-99 | 14 | 10533 | 62 | 10470 | 231 |
| 7A-6 | 15-843 | 16-Jan-99 | 14 | 8253 | 103 | 8151 | 322 |
| 7A-7 | 15-844 | 30-Jan-99 | 14 | 11344 | 140 | 11204 | 321 |
| 7A-8 | 15-845 | 13-Feb-99 | 14 | 7691 | 48 | 7643 | 322 |
| 7A-9 | 15-846 | 27-Feb-99 | 14 | 5866 | 196 | 5670 | 330 |
| 7A-10 | 15-847 | 13-Mar-99 | 14 | 12146 | 324 | 11822 | 337 |
| 7A-11 | 15-848 | 27-Mar-99 | 14 | 29318 | 588 | 28730 | 349 |
| 7A-12 | 15-849 | 10-Apr-99 | 14 | 35024 | 916 | 34108 | 344 |
| 7A-13 | 15-850 | 24-Apr-99 | 9 | 31754 | 1452 | 30205 | 328 |

**Table S2.** Measured 'physico-chemical' and 'biological' variables used in RDA$_{envi}$ and RDA$_{biol}$ multivariate analyses. The first number in the USC sample ID is the deployment (3 through 7), the letter is the trap (Trap A), and the last number is the cup (1 through 13). Precipitation data were acquired from the GPCP Version 1.2 One-Degree Daily Precipitation Data Set (data set ds728.3, https://rda.ucar.edu). Data from the 'Trap' and 'Water column' data sets were obtained from the Cariaco Ocean Time Series Program (available at www.imars.usf.edu/cariaco). Water column data correspond to measurements at 25 m depth, except for Chl *a* data that was integrated over the upper 100 m of the water column. 'Photopigment' data were obtained from Pinckney et al. (2015) and were integrated over the upper 100 m. 'Palynological' data were generated in this study, and the 'Siliceous' data set was obtained from Romero et al. (2009). The reader is referred to Sections 3.1 and 3.4 of the article for more details.